# Quantitative imaging of RAD51 expression as a marker of platinum resistance in ovarian cancer

Michal M Hoppe[1], Patrick Jaynes[1], Joanna D Wardyn[1], Sai Srinivas Upadhyayula[1], Tuan Zea Tan[1], Stefanus Lie[1], Diana G Z Lim[2], Brendan N K Pang[1,2], Sherlly Lim[1], Joe P S Yeong[1], Anthony Karnezis[3,†], Derek S Chiu[3], Samuel Leung[3], David G Huntsman[3], Anna S Sedukhina[4], Ko Sato[4], Monique D Topp[5], Clare L Scott[5], Hyungwon Choi[6], Naina R Patel[7], Robert Brown[7], Stan B Kaye[8], Jason J Pitt[1], David S P Tan[1,8,*] & Anand D Jeyasekharan[1,8,**]

## Abstract

Early relapse after platinum chemotherapy in epithelial ovarian cancer (EOC) portends poor survival. *A-priori* identification of platinum resistance is therefore crucial to improve on standard first-line carboplatin–paclitaxel treatment. The DNA repair pathway homologous recombination (HR) repairs platinum-induced damage, and the HR recombinase RAD51 is overexpressed in cancer. We therefore designed a REMARK-compliant study of pre-treatment RAD51 expression in EOC, using fluorescent quantitative immunohistochemistry (qIHC) to overcome challenges in quantitation of protein expression *in situ*. In a discovery cohort (n = 284), RAD51-High tumours had shorter progression-free and overall survival compared to RAD51-Low cases in univariate and multivariate analyses. The association of RAD51 with relapse/survival was validated in a carboplatin monotherapy SCOTROC4 clinical trial cohort (n = 264) and was predominantly noted in HR-proficient cancers (Myriad HRDscore < 42). Interestingly, overexpression of RAD51 modified expression of immune-regulatory pathways *in vitro*, while RAD51-High tumours showed exclusion of cytotoxic T cells *in situ*. Our findings highlight RAD51 expression as a determinant of platinum resistance and suggest possible roles for therapy to overcome immune exclusion in RAD51-High EOC. The qIHC approach is generalizable to other proteins with a continuum instead of discrete/bimodal expression.

**Keywords** HRD; immune exclusion; multiplexed IHC; ovarian cancer; RAD51
**Subject Categories** Biomarkers; Cancer

See also: J Schwickert et al (May 2021)

## Introduction

Epithelial ovarian cancer (EOC) is the most lethal of all female genital tract cancers. Platinum chemotherapy is the cornerstone of treatment for EOC, typically combined with paclitaxel. The duration of disease control after platinum chemotherapy is a strong predictor of overall survival in EOC (Davis *et al*, 2014). In recurrent EOC, the platinum treatment-free interval strongly correlates with subsequent response to platinum rechallenge therapy (Markman *et al*, 1991). Platinum resistance (defined as relapse within six months following completion of platinum chemotherapy) occurs in 20–30% of cases. However, recent consensus guidelines highlight the predictive limitations of this "time-based" definition (Colombo *et al*, 2019). There remains a need to *a-priori* identify patients who will have platinum resistance, and there are no molecular markers of platinum resistance in current clinical use. The identification of cases for whom first-line carboplatin–paclitaxel chemotherapy is sub-optimal will facilitate trials of early incorporation of novel agents to improve overall survival.

The sensitivity of ovarian cancers to platinum chemotherapy is in part due to a high prevalence of aberrations in the DNA repair pathway of homologous recombination (HR; McMullen *et al*, 2020). Platinum treatment leads to inter-strand cross-links, which are typically repaired by the pathways of nucleotide excision repair (NER) and HR (De Silva *et al*, 2000; Sarkar *et al*, 2006). HR deficiency (HRD) e.g., with *BRCA* mutations, is associated with exquisite

1 Cancer Science Institute of Singapore, National University of Singapore, Singapore
2 Department of Pathology, National University Hospital, Singapore
3 British Columbia Cancer Agency, Vancouver, BC, Canada
4 Department of Pharmacogenomics, St. Marianna University, Kawasaki, Japan
5 The Walter and Eliza Hall Institute of Medical Research, Parkville, Vic., Australia
6 Saw Swee Hock School of Public Health, National University of Singapore, Singapore
7 Division of Cancer, Imperial College London, London, UK
8 Department of Haematology-Oncology, National University Hospital, Singapore
*Corresponding author. Tel: +65 6773 7888; E-mail: david_sp_tan@nuhs.edu.sg
**Corresponding author. Tel: +65 6516 5094; E-mail: csiadj@nus.edu.sg
†Present address: Pathology and Lab medicine, UC Davis Medical Centre, Sacramento, CA, USA

platinum sensitivity due to the inability to repair platinum cross-links (Tan *et al*, 2008). Up to 50% of EOC show HRD through mutations in other HR regulatory genes of the *BRCA*/Fanconi Anaemia (FA) network (Bell *et al*, 2011). However, unlike other *BRCA*/FA genes, RAD51—the central recombinase of the HR pathway—is not commonly mutated in cancer. Depletion or mutation of RAD51 is lethal due to its essential role in cellular replication (Tsuzuki *et al*, 1996; Sonoda *et al*, 1998). Conversely, RAD51 is often upregulated in multiple cancer types and is associated with poor survival (Qiao *et al*, 2005; Mitra *et al*, 2009; Tennstedt *et al*, 2013; Alshareeda *et al*, 2016). As a corollary to platinum sensitivity in ovarian cancers with HRD, it is not known if the overexpression of RAD51 confers platinum resistance. However, evaluating the clinical significance of RAD51 overexpression has been hampered by the lack of quantitative tools for proteomics *in situ*.

In this paper, we utilize quantitative immunohistochemistry (qIHC) through multispectral imaging/ automated analysis to evaluate baseline RAD51 protein levels in formalin-fixed paraffin-embedded (FFPE) tissue. For the discovery cohort, we focused on high-grade serous ovarian carcinomas (HGSOC), the most common and aggressive subtype of EOC. Platinum is typically combined with paclitaxel, the sensitivity to which is not associated with HRD, making it challenging to dissect the contribution of a biomarker to platinum-specific survival. We therefore validated our findings in samples from the SCOTROC4 clinical trial in a REMARK-compliant retrospective biomarker analysis. SCOTROC4 was a phase III trial of carboplatin monotherapy in EOC, assessing two different dosing schedules (Banerjee *et al*, 2013). While the trial showed no difference between the two arms, it represents a unique cohort of platinum monotherapy in ovarian cancer, with well-annotated survival data and HRD scores (Stronach *et al*, 2018).

## Results and Discussion

RAD51 forms discrete nuclear foci upon activation of HR, and this is a widely used measure of recombination proficiency *in vitro* (Graeser *et al*, 2010). The RAD51 foci counting assay has been evaluated in FFPE and *ex vivo* samples (Graeser *et al*, 2010; Naipal *et al*, 2014; Castroviejo-Bermejo *et al*, 2018; Tumiati *et al*, 2018). However, automated quantitation of foci counts in FFPE and *ex vivo* samples is logistically complex and highly reliant on sample preparation/microscopy setup. Conversely, the quantitation of mean nuclear intensity (nuclear expression score) by qIHC is relatively amenable to automated quantitation and scalability in large data sets. To setup our RAD51 qIHC assay, we first validated a rabbit monoclonal antibody EPR4030(3) (Abcam)—demonstrating specific detection of RAD51 in FFPE cell blocks, *ex vivo* irradiated patient-derived xenografts and control human tissues (Fig EV1A–D). We define RAD51 nuclear expression score ($RAD51_{NES}$) as the average intensity of RAD51 expression measured by qIHC across all imaged tumour cells for a given sample (Fig 1A). In a training cohort of EOC cases ($n = 52$), $RAD51_{NES}$ showed strong concordance with RAD51 H-Scores obtained independently from two board-certified pathologists (Fig 1B). We evaluated RAD51 expression in a HGSOC cohort of cases treated with standard-of-care protocols at British Columbia Cancer (BCC) Vancouver. We observed that the $RAD51_{NES}$ in this cohort followed a normal distribution (Fig 1C). Unlike markers such

as Ki67 or ER/PR which have a distinct bimodal pattern of expression (i.e. a cell is either "positive" or "negative"), many cancer-related proteins display homogenous expression within a sample and normal distribution across samples. To cater for the normal distribution of RAD51 within a clinical cohort, $RAD51_{NES}$ was analysed as either a continuous variable or a categorical variable dividing the cohort into RAD51-Low ($RAD51_{NES}$ first quartile [Q1]), RAD51-High (fourth quartile [Q4]) and RAD51-IQR (IQR [quartiles 2 + 3]). We subsequently applied our optimized protocol for staining, imaging, scoring and analysis to assess the clinical relevance of RAD51 protein expression in the BCC cohort. In a Kaplan–Meier survival analysis, high $RAD51_{NES}$ was associated with poorer progression-free survival (PFS) and overall survival (OS) (Fig 1D). We used the 12-month PFS rate (%) as a surrogate for early relapse after completion of platinum-based chemotherapy. RAD51-High cases showed higher likelihood of progression than RAD51-Low cases at both 12 and 24 months (Fig 1E), pointing to the potential utility of RAD51 in predicting platinum resistance. As RAD51 expression is linked to proliferation through common regulatory pathways (Fischer *et al*, 2016), a possible explanation for poor survival could be increased proliferation in RAD51-High tumours. We therefore measured the proliferation marker Ki67 in the BCC cohort by qIHC. $RAD51_{NES}$ correlated weakly with Ki67 extent (Fig EV2A). Importantly, the proliferation status of the tumour (i.e. extent of Ki67 positivity) was not associated with survival outcomes (Fig EV2B). Furthermore, in a multivariate cox proportional hazards model (Cox PH) adjusting for Ki67 extent, age and stage, $RAD51_{NES}$ as a continuous variable remained a statistically significant independent predictor of PFS in HGSOC. Comparable results were obtained for OS (Table 1).

Platinum is typically used along with paclitaxel in frontline chemotherapy of ovarian cancer. To negate potential confounding effects of paclitaxel on survival outcomes, we utilized the unique carboplatin monotherapy SCOTROC4 trial as a validation cohort for our findings. RAD51 protein expression within this cohort also followed a normal distribution (Fig 2A). RAD51-High patients again showed poorer PFS and OS after platinum monotherapy in comparison to RAD51-Low cases (Fig 2B). In a Cox PH multivariate analysis of continuous $RAD51_{NES}$ and Ki67 extent controlling for clinical prognosticators (Table 2), $RAD51_{NES}$ was not independently associated with poor PFS, but remained an independent statistically significant predictor of OS. Ki67 extent was not significantly associated with PFS or OS in multivariate analyses (Fig EV2CandD, Table 2). We then used PFS rate (%) at 12 months (calculated from time of randomization) as a surrogate for a shorter platinum-free interval and hence platinum resistance. Similar to the BCC cohort, RAD51-High cases were more likely to relapse within both 12 months and 24 months than RAD51-Low cases (Fig 2C). Overall, in two independent cohorts, a high $RAD51_{NES}$ associates with early relapses after platinum treatment in ovarian cancer and implies a higher risk of primary platinum resistance in RAD51-High tumours.

HR deficiency (HRD) is common in ovarian cancer, and cases with HRD are sensitive to platinum and PARP inhibitors (PARPi). The Myriad Genetics genomic scar HRD score was available for 240 patients in the SCOTROC4 cohort and 67 (27.9%) were defined as HRD-positive based on a validated cut-off of $\geq 42$ (Mirza *et al*, 2016; Telli *et al*, 2016). We have previously shown that SCOTROC4 cases with HRD have longer PFS and OS on platinum treatment compared

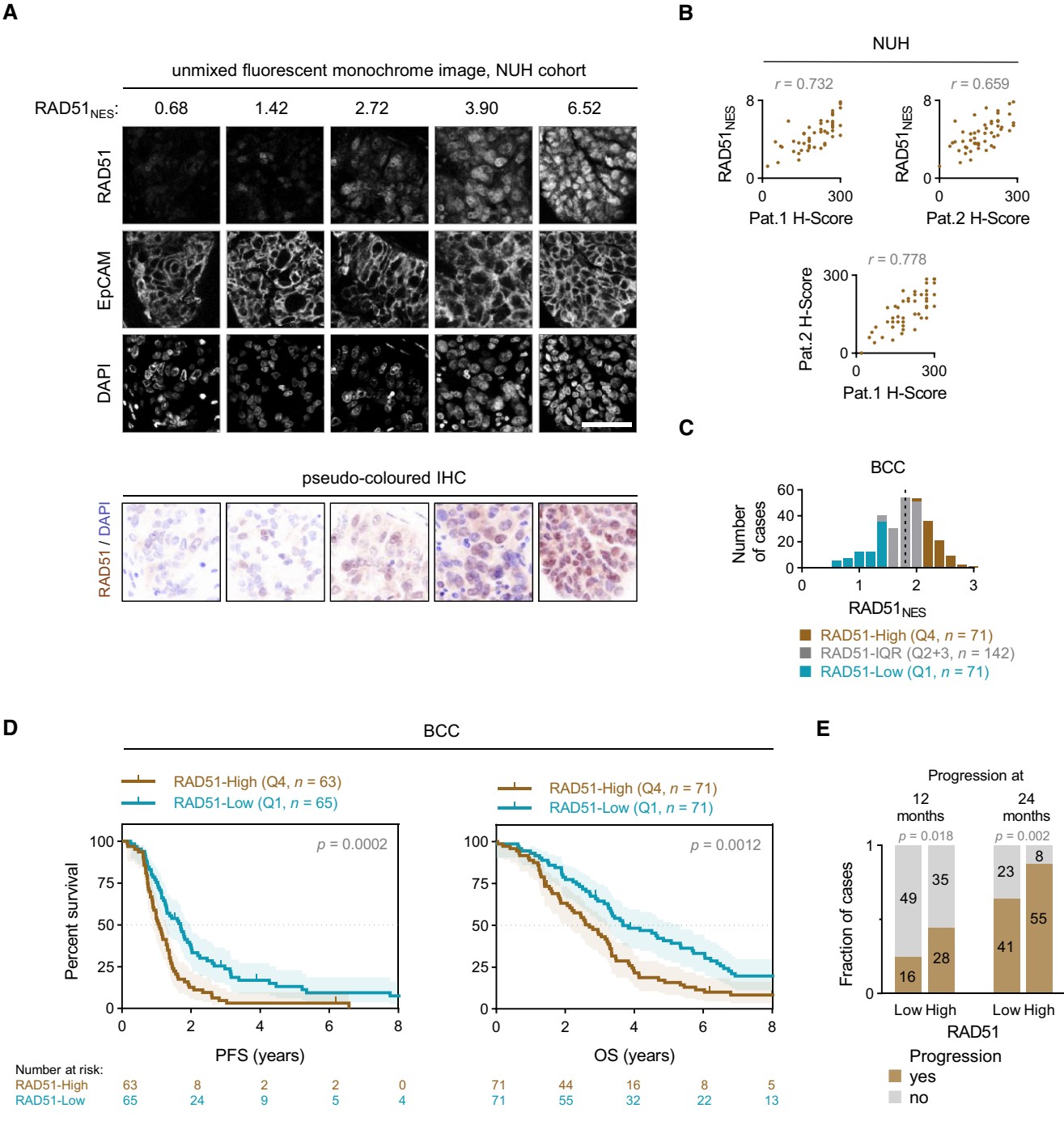

**Figure 1. RAD51 assay development and testing on BCC cohort.**

A　Range of example RAD51 nuclear expression score ($RAD51_{NES}$) values with respective unmixed monochrome fluorescent IHC staining images. EpCAM is used as a tumour marker and an internal sample quality control. Scale bar is 50 µm.

B　Correlation of $RAD51_{NES}$ with two independent pathologist H-scores (*top left* and *top right*) and correlation of two pathologist with each other (*bottom*). Pearson correlation.

C　Distribution of RAD51 nuclear expression score ($RAD51_{NES}$) in the BCC cohort. The cohort is divided into RAD51-Low expressing cases (first quartile, Q1—blue), intermediate cases (interquartile range, IQR—grey) and RAD51-High expressing cases (fourth quartile, Q4—brown). Dashed line denotes the median $RAD51_{NES}$ in this cohort.

D　Survival analysis of the BCC cohort. Kaplan–Meier plots for progression-free survival (PFS) (*left*) and overall survival (OS) (*right*) stratified according to fourth quartile (Q4) and first quartile (Q1) of $RAD51_{NES}$. Log-rank test, shading denotes 95% confidence intervals.

E　Number of cases with progression at 12 and 24 months. Chi-square test.

**Table 1. Multivariate analysis of continuous RAD51$_{NES}$ and Ki67 extent as a predictor of PFS and OS in the BCC cohort of HGSOC (Cox proportional hazards model).**

| Variable | Total cases ($n = 242$) missing values ($n = 43$) | | Total cases ($n = 278$) missing values ($n = 7$) | |
|---|---|---|---|---|
| | PFS | | OS | |
| | HR (95% CI) | P-value | HR (95% CI) | P-value |
| RAD51$_{NES}$ (continuous) | 1.4 (1.0 to 1.9) | 0.025 | 1.3 (0.98 to 1.9) | 0.066 |
| Ki67% (continuous) | 0.97 (0.54 to 1.7) | 0.975 | 0.84 (0.49 to 1.4) | 0.529 |
| Age | | | | |
| <65 | Ref. | | Ref. | |
| ≥65 | 1.0 (0.79 to 1.4) | 0.797 | 1.3 (1.0 to 1.7) | 0.022 |
| Stage | | 0.048 | | 0.011 |
| I | Ref. | | Ref. | |
| II | 1.8 (0.70 to 4.4) | 0.230 | 1.4 (0.53 to 3.7) | 0.500 |
| III | 2.6 (1.2 to 5.6) | 0.015 | 2.7 (1.2 to 6.2) | 0.017 |
| IV | 2.2 (0.9 to 5.3) | 0.086 | 2.3 (0.91 to 5.7) | 0.078 |

BCC, British Columbia Cancer; CI, Confidence interval; HGSOC, High-grade serous ovarian cancer; HR, Hazard ratio; OS, overall survival; PFS, progression-free survival; RAD51$_{NES}$, RAD51 nuclear expression score; Ref., Reference sample.

to HRD-negative patients (Stronach *et al*, 2018). Since cancers with HRD are platinum sensitive and we identify RAD51-High cancers to be platinum resistant, we evaluated the interaction between a genomic scar HRD score (Myriad Genetics) and the RAD51$_{NES}$ in the SCOTROC4 cohort. Neither *BRCA* mutations nor absolute HRD scores associated with RAD51$_{NES}$ (Fig EV2E and F), suggesting that the mechanisms driving RAD51 expression in cancer are unrelated to the presence of a recombination defect. We then performed a subset survival analysis of HRD-positive and -negative tumours in the SCOTROC4 cohort. Interestingly, within the HRD-positive group, the RAD51$_{NES}$ did not stratify patients for survival (Fig 2D) while we observed a clear association of the RAD51$_{NES}$ with survival within the HRD-negative subgroup: RAD51-Low cases showed an increase in PFS and OS compared to RAD51-High cases (Fig 2E). The association of RAD51 with OS remained significant in this HRD-negative group using a multivariate Cox PH analysis (Appendix Table S1). These results suggest that RAD51 expression is most relevant in predicting for platinum resistance within "HRD-negative" i.e., HR intact, EOC cases. Biologically, we speculate that RAD51's role in replication fork protection and reversal requires intactness of other recombination proteins (Mijic *et al*, 2017; Mason *et al*, 2019). Of clinical relevance, HRD negativity is noted in 60–70% of all ovarian cancers (50% of HGSOC), and incorporating RAD51 expression assays alongside HRD assays may help identify cases that are likely platinum resistant.

Despite widespread advances in automated imaging and quantitation, there are still no clinically applied digital pathology assays in oncology. This cohort study of a digital pathology-based protein expression biomarker provides a generalizable roadmap to explore

the clinical relevance of protein expression, the majority of which follow Gaussian distributions in cancer. A limitation of our study is the lack of a RAD51$_{NES}$ "cut-off" which unequivocally denotes resistance to platinum therapy. Defining a cut-off requires a prospective study with standardized protocols optimized for sample preparation and suitable reference standard controls. Furthermore, our protocol necessitates the use of a spectral camera to define and unmix autofluorescence—a common technical drawback of fluorescent IHC. Having identified RAD51 as a key determinant of platinum resistance using the multispectral method, a future comparison of methods for quantitative measurement (e.g. digital spatial profiling/ non-spectral qIHC) should evaluate which would be best for a robust determination of a cut-off for clinical use. Ideally, this would be done in prospective cohorts from clinical trials of platinum/taxol and PARPi treatment in EOC, with availability of HRD scores to validate the relevance of the RAD51-High/HRD-negative "subgroup".

As RAD51 overexpression was associated with early relapse after platinum treatment in EOC, we aimed to evaluate other phenotypic associations with RAD51-High, towards defining possible treatment strategies for these cancers. We first created a set of RAD51 overexpression HGSOC cell lines (Domcke *et al*, 2013) (Fig EV3A). The overexpressed construct was functional, as evidenced by its ability to rescue the loss of cell viability under conditions of RAD51 depletion (Fig EV3B) and formation of foci after DNA damage induction (Fig EV3C). However, we did not observe increased platinum resistance upon RAD51 overexpression *in vitro* using viability and growth assays (Fig 3A and B). We then performed a transcriptomic analysis comparing RAD51 overexpressing and control HGSOC cell lines ($n = 4$). Interestingly, even though there were no highly enriched single transcripts, gene set enrichment analysis (GSEA) showed that RAD51-overexpression cell lines showed overall enrichment in genes related to regulation of T-cell- and B-cell-mediated immunity (Fig 3C, Tables EV1 and EV2). We postulate that altered levels of RAD51 could affect aberrant generation of immunogenic nucleic acids from replication fork structures (Bhattacharya *et al*, 2017), which could in turn affect prognosis through effects on the immune microenvironment of RAD51-High cancers (Bever & Le, 2018). Corroborating this, in clinical samples of the TCGA HGSOC cohort, among the pathways highly associated with RAD51 were interferon responses (Fig 3D, Table EV3). We then interrogated a curated set of immune-related genes with RAD51 in four distinct mRNA cohorts of EOC (TCGA, AOCS, MGH, Duke) and identified a remarkably consistent enrichment of specific immune genes in RAD51-High tumours (Figs 3E and F, and EV4; Table EV4). These included consistent upregulation of genes regulating antigen presentation (TFRC, PSMB7/9, TAP1/2) and chemokines (CXCL10, CCL8), but also of certain negative immune checkpoints such as CD47. We therefore aimed to further define the immune microenvironment of RAD51-High cancers at single-cell resolution using multispectral qIHC. We focused on T cells and immunosuppressive macrophages due to their known prognostic role in EOC, using the immune markers CD3, CD8, FOXP3 and CD163 along with cytokeratin as a tumour mask to separate stromal and tumour compartments (Fig EV5A). Using the platinum–taxol-treated BCC cohort, we observed a significant exclusion of CD3$^+$/CD8$^+$ cytotoxic T cells from the tumour regions in RAD51-High cancers (Fig 3G and H). A similar but less prominent effect was noted with the total T-cell population

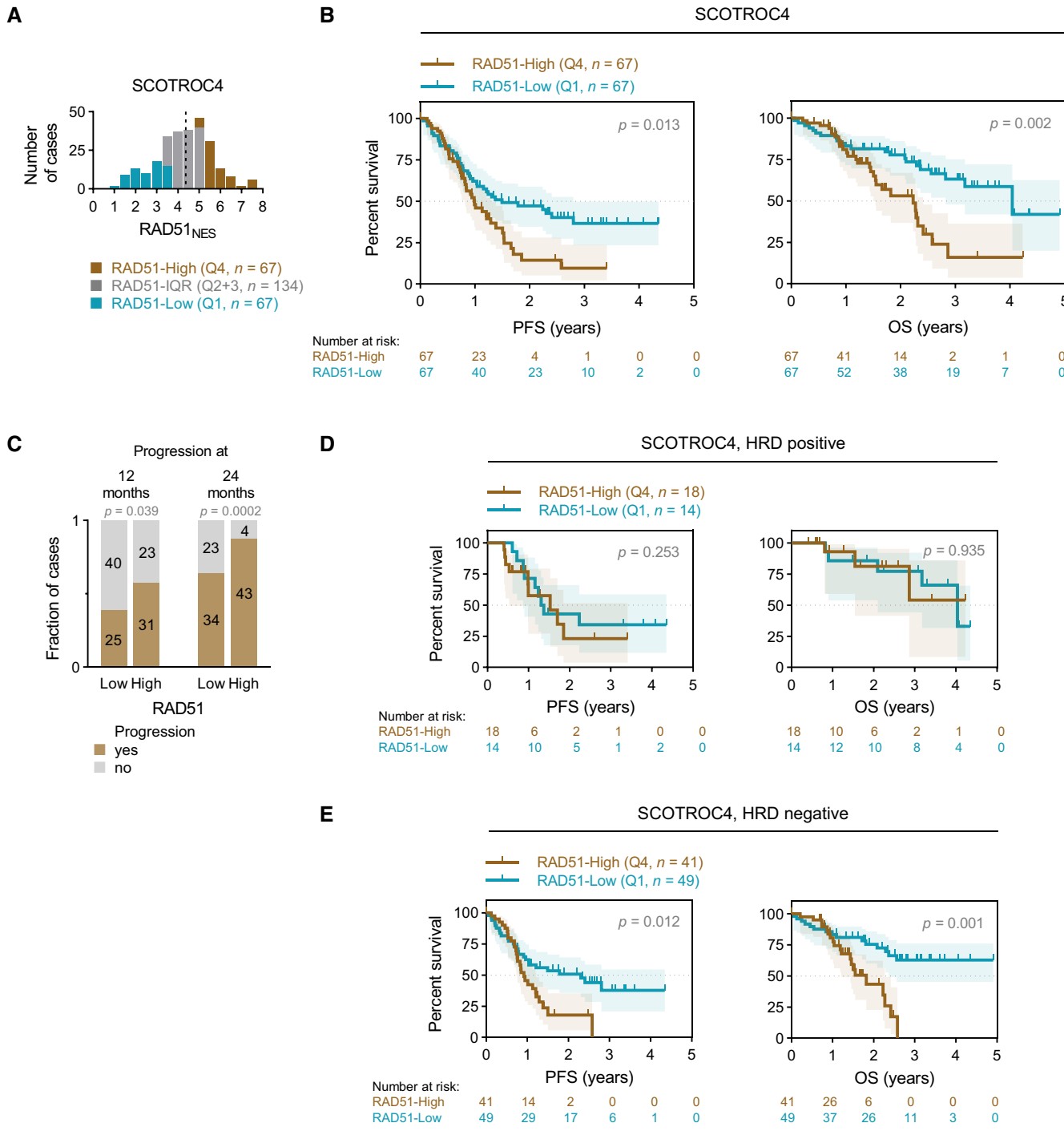

**Figure 2. Validation of assay and findings in SCOTROC4 cohort.**

A  Distribution of RAD51$_{NES}$ in the SCOTROC4 cohort. Dashed line denotes the median RAD51$_{NES}$ in this cohort.
B  Survival analysis of the SCOTROC4 cohort. Kaplan–Meier plots for PFS (*left*) and OS (*right*) stratified according to quartiles of RAD51$_{NES}$. Log-rank test, shading denotes 95% confidence intervals.
C  Number of cases with progression at 12 and 24 months. Chi-square test.
D  Survival analysis of HRD-positive patients according to quartile of RAD51$_{NES}$. Log-rank test, shading denotes 95% confidence intervals.
E  Survival analysis of HRD-negative patients. Log-rank test, shading denotes 95% confidence intervals.

(CD3$^+$ only) and CD3$^+$/FOXP3$^+$ T regulatory cells, but not with CD163$^+$ macrophages (Fig EV5B). Our results mirror prior work in lung cancer, where low RAD51 was associated with increased TILs

(Gachechiladze *et al*, 2020). The cytotoxic T-cell exclusion phenotype in our RAD51-High cases was primarily noted in *BRCA* wild-type (WT) tumours (Fig EV5C), in keeping with the prognostic

**Table 2. Multivariate analysis of continuous RAD51$_{NES}$ and Ki67 extent as a predictor of PFS and OS in the SCOTROC4 cohort (Cox proportional hazards model).**

| Variable | Total cases ($n = 175$) missing values ($n = 93$) | | | |
| --- | --- | --- | --- | --- |
| | PFS | | OS | |
| | HR (95% CI) | *P*-value | HR (95% CI) | *P*-value |
| RAD51$_{NES}$ (continuous) | 1.2 (0.97 to 1.5) | 0.104 | 1.4 (1.1 to 1.9) | 0.007 |
| Ki67% (continuous) | 1.0 (0.98 to 1.02) | 0.971 | 0.98 (0.96 to 1.0) | 0.227 |
| Age | | | | |
| <65 | Ref. | | Ref. | |
| ≥65 | 1.0 (0.64 to 1.6) | 0.996 | 0.99 (0.54 to 1.8) | 0.977 |
| Stage | | <0.001 | | 0.023 |
| I | Ref. | | Ref. | |
| II | 3.8 (0.99 to 15.0) | 0.052 | 13.7 (1.6 to 120.5) | 0.018 |
| III | 9.7 (2.8 to 33.1) | <0.001 | 11.2 (1.4 to 88.9) | 0.022 |
| IV | 5.3 (1.4 to 20.2) | 0.014 | 4.8 (0.54 to 44.0) | 0.160 |
| Histology | | 0.150 | | <0.001 |
| Serous | Ref. | | Ref. | |
| Mucinous | 3.9 (1.3 to 12.4) | 0.019 | 16.3 (4.4 to 60.9) | <0.001 |
| Clear cell | 2.4 (0.63 to 8.9) | 0.205 | 16.3 (3.7 to 72.3) | <0.001 |
| Endometrioid | 0.94 (0.43 to 2.0) | 0.866 | 1.1 (0.36 to 3.3) | 0.879 |
| Other | 1.7 (0.38 to 8.0) | 0.473 | 0.60 (0.07 to 5.0) | 0.636 |
| Grade (differentiation) | | | | |
| 1—well | Ref. | | Ref. | |
| 2—moderate and 3—poor | 2.9 (1.0 to 8.0) | 0.042 | 7.5 (0.98 to 57.9) | 0.053 |
| Performance status | | 0.082 | | 0.032 |
| 0 | Ref. | | Ref. | |
| 1 | 2.0 (1.0 to 3.8) | 0.508 | 1.4 (0.66 to 3.0) | 0.372 |
| 2 and 3 | 1.2 (0.71 to 2.0) | 0.037 | 3.1 (1.2 to 7.9) | 0.017 |
| Bulk of residual disease | | 0.003 | | 0.035 |
| None/microscopic | Ref. | | Ref. | |
| Macroscopic < 2 cm | 2.5 (1.5 to 4.2) | 0.001 | 2.7 (1.3 to 5.7) | 0.010 |
| Macroscopic > 2 cm | 2.0 (1.1 to 3.6) | 0.027 | 2.2 (0.95 to 5.1) | 0.065 |
| HRD score | | | | |
| HRD-positive | Ref. | | Ref. | |
| HRD-negative | 1.9 (1.2 to 3.1) | 0.005 | 2.3 (1.23 to 4.3) | 0.009 |

CI, Confidence interval; HR, Hazard ratio; HRD, Homologous Recombination Deficiency score; OS, overall survival; PFS, progression-free survival; RAD51$_{NES}$, RAD51 nuclear expression score; Ref., Reference sample.

significance of RAD51 in HRD-negative patients, who are typically *BRCA* WT (Fig 2E). We speculate that RAD51-High promotes an as yet unknown immune checkpoint that prevents T-cell infiltration (predominantly cytotoxic T cells, but also other T-cell subsets) into the tumour from the stroma. The elevated CXCL chemotactic signals and expression of antigen-presenting genes may represent an "ineffective" compensation to this negative checkpoint, ultimately resulting in evasion of immune surveillance and poor survival. These results also point to possible therapeutic approaches for RAD51-High tumours e.g., the addition of CTLA4 inhibitors to promote T-cell recruitment. The consistent correlation of CD47 expression with RAD51 is also interesting, and further work will be required to understand its biological significance in this setting along with the potential applicability of anti-CD47 monoclonal antibodies. Finally, given the proven clinical utility of anti-angiogenic drugs (e.g. Bevacizumab) in EOC, it will be interesting to evaluate whether their role in remodelling tumour vasculature to facilitate T-cell migration (Dickson *et al*, 2007; Wallin *et al*, 2016) can specifically overcome the platinum resistance seen in RAD51-High EOC.

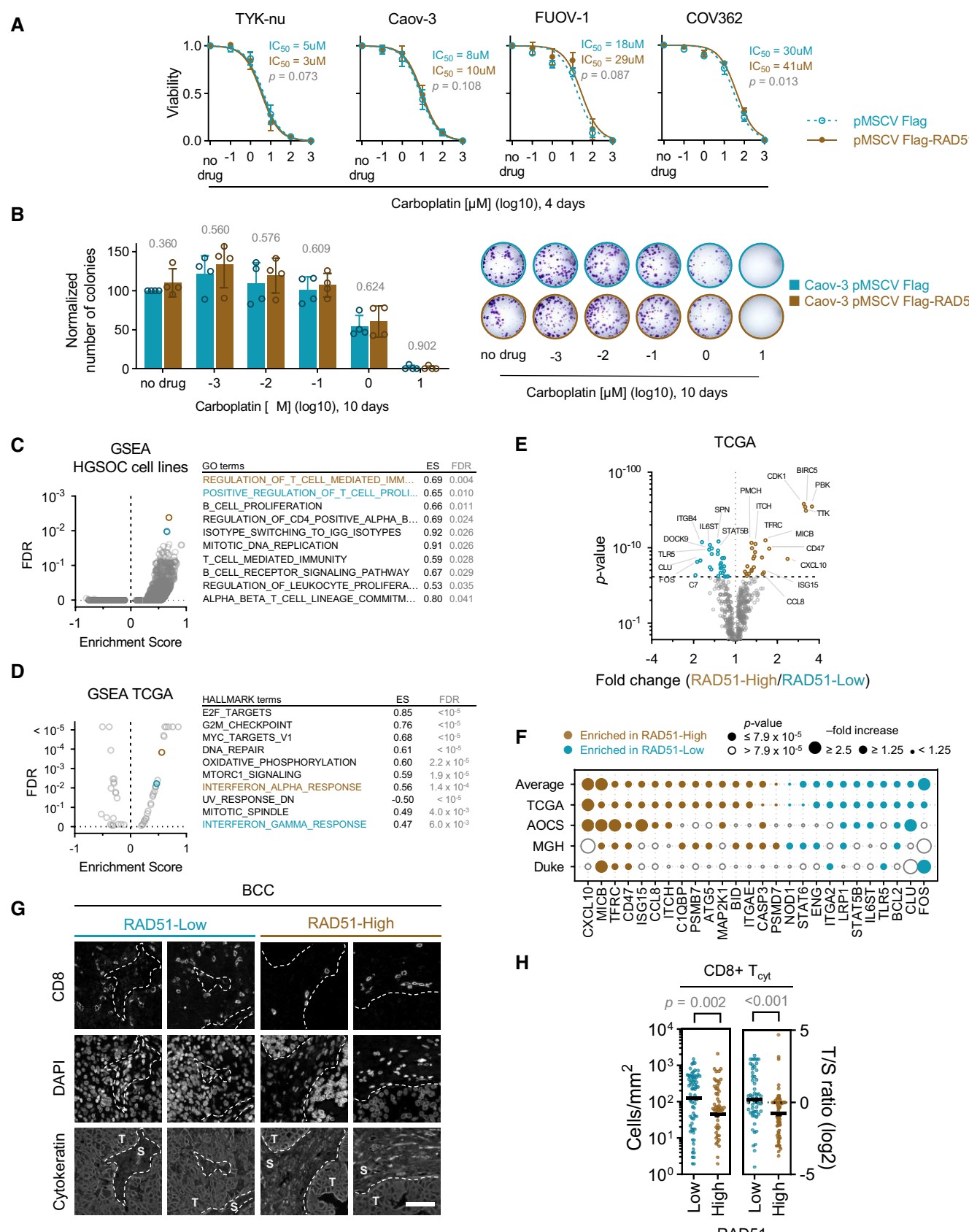

**Figure 3.**

**Figure 3.  RAD51 does not promote platinum resistance *in vitro* but modulates immune-related gene expression.**

A  *In vitro* cell survival assay of HGSOC cell lines upon stable RAD51 overexpression. Mean with standard deviation is shown of at least three biological replicates per point. Extra-sum-of-squares *F*-test.

B  *In vitro* colony-forming assays comparing RAD51-overexpressing and control HGSOC cell line, Caov-3, after treatment with increasing doses of carboplatin. Mean and SD of four biological replicates (*left*) a representative experiment (*right*). *P*-value for a comparison between cell lines for each drug treatment condition is indicated above the bars. *t*-Test.

C  Gene Set Enrichment Analysis of RAD51-overexpression vs. control HGSOC cell lines ($n = 4$). The top ten enriched pathways in RAD51-overexpression cell lines are listed on the right, ranked by the lowest false-discovery rate (FDR). The corresponding Enrichment Score (ES) is shown. See Table EV1 for details on all data points. Colouring denotes the top two enriched pathways. GO—Gene Ontology.

D  Gene Set Enrichment Analysis of RAD51-High vs. RAD51-Low HGSOC tumours from a TCGA cohort. The top ten enriched pathways in RAD51-High tumours are listed on the right, ranked by the lowest false-discovery rate (FDR). See Table EV3 for details on all data points. Colouring denotes genesets related to immune response pathways. ES—Enrichment Score.

E  Volcano plot for fold changes of immune genes enriched or depleted in RAD51-High tumours vs. RAD51-Low tumours from TCGA. *t*-Test; dashed line denoted threshold of significance, Bonferroni corrected for multiple testing.

F  Immune genes enriched in RAD51-High and RAD51-Low tumours across four EOC mRNA cohorts (TCGA, AOCS, MGH, Duke).

G  Example unmixed fluorescent IHC images demonstrating the presence of CD8[+] T cells in the tumour (T) and stroma (S) in a RAD51-High and a RAD51-Low tumour from the BCC cohort. Scale bar is 50 μm.

H  CD8[+] cytotoxic T-cell ($T_{cyt}$) infiltration analysis in the BCC cohort. Absolute tumour cytotoxic T-cell infiltration numbers and tumour/stroma (T/S) cytotoxic T-cell number ratio in RAD51-High and -Low cases (*left*). Bar is median. Mann–Whitney test.

# Materials and Methods

## Patients and treatment

EOC FFPE samples to optimize RAD51 fluorescent quantitative immunohistochemistry (qIHC) were obtained from National University Hospital of Singapore (NUH cohort, $n = 52$). A cohort of HGSOC was obtained from British Columbia Cancer (BCC TMA, $n = 308$, Appendix Table S2). Samples of EOC from the SCOTROC4 clinical trial ($n = 309$, Appendix Table S3), chosen based on sample availability and divided equally between both study arms (Banerjee *et al*, 2013), were used as a tissue microarray (TMA). HRD score from the Myriad genomic scar assay was available for 240/309 cases on this TMA (Stronach *et al*, 2018). The NUH cohort study was approved by the NHG DSRB (Ref #: 2014/00989); the BCC HGSOC cohort study was approved by BC Cancer (REB #: H05-60119); and the SCOTROC4 cohort study was approved by the UK ethics committee. In all studies, informed consent, or waiver of consent, was obtained from all subjects and the experiments conformed to the principles set out in the WMA Declaration of Helsinki and the Department of Health and Human Services Belmont Report. For information pertaining to patients and ethics for samples used for patient-derived xenografts, see Topp *et al* (2014). Approval to analyse samples from the above cohorts came from the NUS IRB (Ref #: H-19-055E).

## Fluorescent quantitative immunohistochemistry (qIHC)

Multiplexed qIHC was performed on FFPE samples to assess protein expression using an Opal 7-Color Kit and imaged/analysed using the Vectra 2 System (PerkinElmer Inc., Waltham, MA, USA). The RAD51 Vectra score ($RAD51_{NES}$) measures mean nuclear fluorescent intensity per tumour expressed in normalized counts. The results of this study are reported in concordance with REMARK guidelines (Sauerbrei *et al*, 2018).

Briefly, 3 μm thick tissue sections were deparaffinized in organic solvents and rehydrated using a gradient of ethanol solutions. Heat-mediated antigen retrieval was performed on rehydrated slides using Target Retrieval solutions (Dako, Denmark), followed by

30 min blocking with Dako Ab Diluent (Dako, Denmark) and 1h primary antibody incubation at room temperature. Primary antibodies used in this study are listed in Appendix Table S4. Slides were washed in 0.1% Tween20 (Sigma-Aldrich) solution in water 3 times, 2 min each wash with agitation. Secondary antibody incubation was 10 min at room temperature using anti-mouse or anti-rabbit IgG HRP-labelled secondary antibodies (1:1,000, PerkinElmer Inc., Waltham, MA, USA) followed by a washing cycle. Finally, slides were incubated with choice of Opal fluorophore (1:100) for 5 min at room temperature, followed by a wash cycle. This constitutes a full cycle of antibody staining and can be multiplexed by repeating this sequence starting from heat-mediated antigen retrieval using a microwave to strip the specimen of antibodies present from the previous round. DAPI was added to the final secondary antibody mixture to serve as a counterstain. Slides were mounted using Mowiol 4-88-based mounting medium (Sigma-Aldrich).

Slides were imaged using the Multispectral Vectra 2 Imaging System (PerkinElmer Inc., Waltham, MA, USA). Multispectral images were analysed using the inForm 2.2 software (PerkinElmer Inc., Waltham, MA, USA). To obtain monochrome images of all components of a multiplexed qIHC slide, images were first unmixed using a prepared spectral library of fluorophore-specific pure spectra measured from single stained slides of each fluorophore. EOC-specific autofluorescence spectra were also obtained from an unstained, but processed slide. Next, a trainable tissue segmentation algorithm was used to identify regions of interest in images i.e., epithelial tumour cells, but not stromal cell. With each multiplexed qIHC, we included a staining against EpCAM or cytokeratin, markers of epithelial cells. Correct tissue segmentation was reviewed to ensure reliability of the segmentation protocol. All cells included in regions positively stained with the anti-EpCAM/cytokeratin antibody were included in the tumour-specific cell-based analysis. Cells were segmented using the cell segmentation algorithm, creating a nuclear mask for each cell within the tumour region. The nuclear mask was based on the counterstain and correct segmentation was aided by the presence of the EpCAM/cytokeratin membrane marker to help define cellular boundaries. Mean nuclear fluorescent intensity of a marker of

interest was denoted as an expression score and used as a read-out of protein expression. The score is expressed in Normalized Counts as follows:

$$\text{Normalized Counts} = \frac{\text{fluorescent counts}}{2^{\text{bit depth}} \times \text{exposure time} \times \text{gain} \times \text{binning area}}$$

Exposure time is expressed in seconds, and binning area is 1 for all images. TMA cores with < 100 tumour cells, damaged tissue or unwarranted staining patterns were not included in further analysis and were considered as failed quality control. In the SCOTROC4 cohort, median number of cells analysed per TMA core was 1596 (range 124–4,590) and at least two TMA cores were analysed per patient (range 2–12). Routine biomarker evaluation centres on group stratification across the median value. However, as $\text{RAD51}_{\text{NES}}$ follows a normal distribution within the cohort (Figs 1C and 2A), stratification across the median is non-optimal. Division across median dichotomizes the peak cases into High and Low groups, saturating them with cases of quantitatively similar results. To preserve the biological distinction relating to RAD51 expression in a normally distributed cohort, we reasoned that stratifying the cohort into three biologically distinct groups will reveal truer associations between RAD51 and survival. Thus, we divided cohorts into RAD51-Low cases within the first quartile (Q1), RAD51-High cases within the fourth quartile (Q4) and RAD51-IQR cases within the interquartile range (IQR, quartiles 2 + 3).

## Statistical methods

For categorical analyses, quantitative scores were divided into three groups accordingly—first quartile (Q1), interquartile range (IQR) and fourth quartile (Q4). For the BCC cohort, progression-free survival (PFS) and overall survival (OS) were defined as time from the date of diagnosis to progression or death, respectively. For the SCOTROC4 data, PFS and OS were defined as time from the date of randomization to progression (PFS) or death (OS) and estimated using the Kaplan–Meier method. Tumour progression was determined according to RECIST version 1.0 criteria. CT scans were carried out at baseline and after six cycles of treatment and if CA125 rose or clinical progression was suspected (Banerjee et al, 2013). Kaplan–Meier curves are shown for Q4 and Q1. Cox proportional hazards (Cox PH) models were calculated using IBM SPSS Statistics 23 software; all variables satisfied the proportional hazards assumption and all available clinically relevant clinicopathological variables were included in the multivariate models. Other statistical tests and graphs were generated using GraphPad Prism 8 software. All assumptions have been met for the use of these statistical tests. Statistical tests were two-sided and $P < 0.050$ was considered as statistically significant in individual testing; Bonferroni correction was applied to pairwise comparisons when more than two groups were present.

## Creation and validation of RAD51 overexpressing cell lines

Four HGSOC cell lines were chosen to perform in vitro experiments based on previous assessment (Domcke et al, 2013): TYK-nu ($TP53_{\text{mut.}}$, $BRCA1/2_{\text{wild-type}}$), Caov-3 ($TP53_{\text{mut.}}$, $BRCA1/2_{\text{wild-type}}$),

COV362 ($TP53_{\text{mut}}$, $BRCA1_{\text{mut.}}$,$BRCA2_{\text{wild-type}}$) and FUOV-1 ($TP53_{\text{mut.}}$, $BRCA1/2_{\text{wild-type}}$). All cell lines were a gift from the laboratory of Dr Ruby Huang at Cancer Science Institute of Singapore. Cell lines were verified to be free of mycoplasma and were authenticated by the Huang laboratory. The coding sequence (CDS) of the canonical RAD51 transcript (NM_002875.4) was amplified from normal fallopian tube cells (FT33 (Karst & Drapkin, 2012), a kind gift from the laboratory of Dr Ronny Drapkin of Dana-Farber Cancer Institute, Harvard Medical School, Boston, Massachusetts, USA) using AAAAGGATCCGCAATGCAGATGCAGCTTGA (forward) and AAAAGCGGCCGCTCAGTCTTTGGCATCTCCCA (reverse) primers and cloned into the BamHI and NotI cloning sites of the pMSCV-puro-Flag retroviral vector. HGSOC cell lines were transduced with virus containing the exogenous Flag-RAD51 CDS or empty vector and cultured in DMEM medium supplemented with 25 mM HEPES, 10% FBS and 1 µg/ml puromycin; expression of exogenous RAD51 was confirmed by western blotting (Fig EV3A).

To confirm functionality of the Flag-tagged exogenous RAD51 construct, cell survival assays were performed. HGSOC cells lines were seeded at low confluency on a 96-well dish in three technical triplicates, 12 h later siRNA transfection was performed. Lipofectamine RNAiMAX (Thermo Fisher) transfection mixture was prepared according to manufacturer's protocol and RNA was added at 20 nM; the following target sequences were used: siControl - ON-TARGETplus Non-targeting Pool (Dharmacon), siRAD51-CDS – CAGAUUGUAUCUGAGGAAA, siRAD51-3'UTR – TCTTCCTGTTGTGACTGCCAGGATA. Twenty-four hours later, cell culture media were changed and carboplatin (Sigma) was added at 1mM concentration and serially diluted. Ninety-six hours after drug addition, MTT solution was added and cells were kept for an extra 2 h at 37°C to allow crystals to form. Next, Stop solution (37% (v/v) N,N-dimethylformamide, 14.2% (m/v) SDS and 2% acetic acid) was added and the plate was incubated on a shaker to allow complete dissolution of violet crystals. Absorbance at 564 nm was read for each well by a Tecan Infinite 200 PRO plate reader. The extent of light absorbance was considered to be proportional to cell density. For cell survival assays shown in Fig 3A, untransfected cells were treated as described above with carboplatin.

## Western blotting

Cells were lysed with Pierce RIPA buffer (Thermo Fisher) supplemented with Halt™ Protease and Phosphatase Inhibitor (Thermo Fisher) on ice for 30 min. Samples were quantified by Pierce™ BCA assay (Thermo Fisher). Samples of at least 20 µg of protein were run on tris-glycine polyacrylamide gel electrophoresis (PAGE) gel. Proteins from the gel were transferred to PDVF membrane (Bio-Rad) overnight at 150 mA at 4°C. Membranes were blocked for at least 30 min in 5% milk/ tris-buffered saline with 0.1% tween (TBST) and incubated in primary antibody (5% milk/TBST) overnight at 4°C. The following day, membrane was incubated with an anti-rabbit HRP-conjugated (GE) secondary antibody for 1 h at room temperature (RT). Chemiluminescent reagents (Millipore) were applied to the membrane, and Western blot image was developed by film developer (Konica Minolta Film Processor) using film (Konica Minolta).

## Colony formation and cell proliferation assays

Colony formation assay: 800 cells were plated in a 6 well plate and treated with the indicated concentration of carboplatin (0–10,000 nM) for 10 days. Post-treatment, cells were washed three times with PBS and stained with crystal violet followed by washing with water to remove excess stain. Images were obtained under light microscope using AxioVision software. Colonies formed were manually counted. Crystal violet stain composition: 0.5 % crystal violet staining solution in 20% methanol in water.

## Mice

Female NOD-scid IL2Rg$^{null}$ (NSG) mice were used. Mice were 6–10 weeks of age at the start of the experiment, when the PDX fragment was transplanted. All animal work was carried out in the Bioservices department at WEHI. NSG mice were bred at the specific pathogen-free WEHI Clive and Vera Ramaciotti laboratories. All experimental mice were housed in an Australian manufactured vented micro-isolator caging system. The health status of the mice was monitored for a range of bacterial, viral and parasitological pathogens.

All experiments involving mice are approved by the WEHI Animal Ethics Committee. Animal research at WEHI follows guidelines from The National Centre for the Replacement, Refinement & Reduction(3Rs) of Animals in Research and is overseen by the Victorian Bureau of Animal Welfare. The use of animals for scientific purposes in Victoria is governed by the Australian Code for the Care and Use of Animals for Scientific Purposes, 8th Edition, 2013.

## Ex vivo treatment of patient-derived xenografts

Using NSG mice, under anaesthesia, a fragment of fresh human EOC tumour was placed either subcutaneously (1–3 mm$^3$) or via the intra-ovarian bursal (< 1 mm$^3$) method. Subcutaneous tumours were assessed by measuring two perpendicular axes using calipers once weekly. Tumour volume was calculated using the formula: $\pi/6 \times$ [larger diameter $\times$ smaller diameter$^2$]. Once the tumour volume reached 0.7 cm$^3$, the mouse was sacrificed and the tumour was harvested. For ex vivo irradiation, tumour fragments were treated with 450 rad γ-irradiation and placed in culture for four hours prior to formalin fixation. Untreated matched tumour fragments from each PDX were used as controls. There were no inclusion/exclusion criteria set for tumour-bearing mice pertaining to this experiment. Samples were then processed by fluorescent qIHC, as described above, and analysed in an un-blinded fashion. There was no exclusion from analysis of any data points arising from irradiated tumour fragments and their respective controls. The results of this study are reported in concordance with ARRIVE guidelines, where applicable (Percie du Sert et al, 2020).

## Gene Set Enrichment Analysis and differential gene expression analysis

Total RNA was isolated from RAD51-overexpression and control HGSOC cell lines using RNeasy Mini Kit (Qiagen) and standard polyA sequencing was done by NovogeneAIT. Identification of differentially expressed genes and Gene Set Enrichment Analysis

**The paper explained**

**Problem**

Platinum chemotherapy is the cornerstone of treatment for epithelial ovarian cancer (EOC). While the typical first-line chemotherapy of Carboplatin + Paclitaxel is highly effective in EOC, a subset of patients are resistant to or relapse early after treatment and have poor overall survival. It would be advantageous to identify these cases prior to initiation of treatment, to facilitate the testing of novel agents that can supplement or even supplant platinum chemotherapy. There are no molecular markers currently used in pathology labs to define possible platinum-resistance, in large part due to challenges in quantitating expression of candidate proteins in tissue sections.

**Results**

We used a state-of-the-art method for simultaneous staining of multiple proteins in tissue sections along with automated microscopy to quantitatively measure RAD51, a DNA repair protein that is important for the resolution of platinum induced damage. We show using two large independent EOC patient cohorts, cases that expressed a high amount of RAD51 relapsed sooner than those expressing a low amount of RAD51. Furthermore, this phenomenon correlates with an exclusion of anti-tumour immune cells from the microenvironment of cancers with RAD51-High.

**Impact**

Our study identifies RAD51 as a bonafide biomarker for increased likelihood for resistance to platinum chemotherapy in ovarian cancer, which can subsequently be developed into a clinical grade assay for routine diagnostic practice. Furthermore, our observation that RAD51 tumours tend to exclude important anti-cancer immune cells sets the stage for developing therapeutic approaches to increase immune infiltration in these cancers.

(using GO geneset library) comparing RAD51-overexpression vs. control cell lines was performed using standard pipelines available publicly on the CSI NGS Portal (An et al, 2020).

To identify differentially expressed immune genes between RAD51-High (fourth quartile, Q4) and -Low (first quartile, Q1) cases, the processed high-grade endometrioid, high-grade serous ovarian cancer gene expression data of TCGA (n = 566) (Bell et al, 2011; Data ref: Bell et al, 2011), Australian Ovarian Cancer Study (AOCS, GSE9891; n = 267) (Tothill et al, 2008; Data ref: Tothill et al, 2008), Massachusetts General Hospital (MGH, GSE26712; n = 185) (Bonome et al, 2008; Data ref: Bonome et al, 2008) and Duke University Hospital (Duke, GSE3149; n = 146) (Bild et al, 2006; Data ref: Bild et al, 2006) were extracted from CSIOVDB (http://csiovdb.mc.ntu.edu.tw/CSIOVDB.html; Tan et al, 2015). Quartiles were calculated independently for each data set. Statistical analyses were conducted using Matlab® R2016b version 9.1.0.960167, statistics and machine learning toolbox version 11.0 (MathWorks; Natick, MA, USA); t-test was applied for identifying differentially expressed genes.

## Data availability

The RNA-seq data from this publication have been deposited to the Gene Expression Omnibus database https://www.ncbi.nlm.nih.gov/geo/query/acc.cgi?acc=GSE166539 and assigned the identifier GSE166539.

Expanded View for this article is available online.

## Acknowledgements

We thank the patients from the cohorts described in this paper, who generously consent to their tissues being analysed for research. We also thank Prof. Ashok Venkitaraman and Prof. George K Chandy for their critical review of the manuscript. This work was supported by the Cancer Science Institute of Singapore, National University of Singapore PhD Graduate Scholarship to MMH, the Singapore Ministry of Health's National Medical Research Council Transition Award (NMRC/TA/0052/2016) to ADJ, the National Research Foundation Singapore and the Singapore Ministry of Education under its Research Centres of Excellence initiative to ADJ, the Singapore Ministry of Health's National Medical Research Council Individual Research Grant (NMRC/CIRG/1400/2014) and Clinician Scientist Award (NMRC/CSA-INV/0016/2017) to DSPT, the CRUK Imperial Centre to NRP, the Imperial Experimental Medicine Centre (SCOTROC4 TMA construction) and the Cancer Research UK and Ovarian Cancer Action to RB.

## Author contributions

Conceptualization (Experimental design): MMH, DSPT and ADJ. Investigation: (Multiplex IHC, *in vitro* and *ex vivo* experiments): MMH, ShL, JPSY, USS, MDT and CLS. Investigation (Pathological review): DGZL and BNKP. Formal analysis (Bioinformatic analysis): JJP, TZT, StL, ASS and KS. Formal analysis (Biostatistics): MMH, HC, DSC and SaL. Project administration (Coordination of sample provision/ethics approvals): RB, NRP, DGZL, AK, SBK, DGH, DSPT and ADJ. Writing – original draft: MMH, JDW, AK, DGH, RB, JJP, DSPT and ADJ. Writing – review and editing: MMH, PJ, DSPT and ADJ.

## Conflict of interest

ADJ; received consultancy fees from Turbine Ltd, AstraZeneca, Janssen and MSD, along with travel funding from Perkin Elmer, and research funding from Janssen. DSPT; honoraria from AstraZeneca, Roche, Bayer, MSD, Merck Serono, Tessa Therapeutics, Novartis, and Genmab and research funding from AstraZeneca, Bayer and Karyopharm. The other co-authors declare that they have no conflict of interest.

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
