## [Review Process File · EMBO Molecular Medicine]

Quantitative imaging of RAD51 expression as a marker of platinum resistance in Ovarian Cancer

Michal Hoppe, Patrick Jaynes, Joanna Wardyn, Sai Upadhyayula, Tuan Zea Tan, Stefanus Lie, Diana Lim, Brendan NK Pang, Sherly Lim, Joe PS Yeong, Anthony Karnezis, Derek Chiu, Samuel Leung, David Huntsman, Anna Sedukhina, Ko Sato, Monique Topp, Clare Scott, Hyungwon Choi, Naina Patel, Robert Brown, Stanley Kaye, Jason Pitt, David Shao Peng Tan, and Anand Jeyasekharan
DOI: 10.15252/emmm.202013366

Corresponding authors: Anand Jeyasekharan (csiadj@nus.edu.sg) , David Shao Peng Tan (david_sp_tan@nuhs.edu.sg)

Review Timeline:

Submission Date:	2nd Sep 20
Editorial Decision:	21st Oct 20
Author Correspondence:	11th Nov 20
Editor Correspondence:	12th Nov 20
Revision Received:	31st Dec 20
Editorial Decision:	21st Jan 21
Revision Received:	4th Feb 21
Accepted:	9th Feb 21

Editor: Lise Roth

Transaction Report:

21st Oct 2020

Dear Dr. Jeyasekharan,

Thank you for the submission of your manuscript to EMBO Molecular Medicine, and please accept my apologies for the delay in getting back to you, which is due to the fact that one referee needed more time to provide his/her report. We have now received feedback from the three reviewers who agreed to evaluate your manuscript.

As you will see from the reports below, the reviewers find that the question addressed by the study is of potential interest. However, they remain unconvinced that some of the major conclusions are sufficiently supported by the data, and think that both clinical significance and mechanistic insight should be further strengthened before further consideration. Alternatively, and following further discussion with the referees, we agree that this manuscript could be turned into a short report, in which case the mechanistic aspect would not need deeper analysis.

If you feel you can satisfactorily address the points listed by the referees, you may wish to submit a revised version of your manuscript. Please attach a covering letter giving details of the way in which you have handled each of the points raised by the referees. A revised manuscript will once again be subject to review. EMBO Molecular Medicine encourages a single round of revision only and therefore, acceptance or rejection of the manuscript will depend on the completeness of your responses included in the next, final version of the manuscript. For this reason, and to save you from any frustrations in the end, I would strongly advise against returning an incomplete revision.

When submitting your revised manuscript, please carefully review the instructions that follow below. Failure to include requested items will delay the evaluation of your revision:

- 1) A .docx formatted version of the manuscript text (including legends for main figures, EV figures and tables). Please make sure that the changes are highlighted to be clearly visible.
- 2) Individual production quality figure files as .eps, .tif, .jpg (one file per figure).
- 3) A .docx formatted letter INCLUDING the reviewers' reports and your detailed point-by-point responses to their comments. As part of the EMBO Press transparent editorial process, the point-by-point response is part of the Review Process File (RPF), which will be published alongside your paper.
- 4) A complete author checklist, which you can download from our author guidelines (<https://www.embopress.org/page/journal/17574684/authorguide#submissionofrevisions>). Please insert information in the checklist that is also reflected in the manuscript. The completed author checklist will also be part of the RPF.
- 5) Before submitting your revision, primary datasets produced in this study need to be deposited in

an appropriate public database (see <https://www.embopress.org/page/journal/17574684/authorguide#dataavailability>). Please remember to provide a reviewer password if the datasets are not yet public. The accession numbers and database should be listed in a formal "Data Availability " section (placed after Materials & Method). Please note that the Data Availability Section is restricted to new primary data that are part of this study.

6) We would also encourage you to include the source data for figure panels that show essential data. Numerical data should be provided as individual .xls or .csv files (including a tab describing the data). For blots or microscopy, uncropped images should be submitted (using a zip archive if multiple images need to be supplied for one panel). Additional information on source data and instruction on how to label the files are available at .

7) Our journal encourages inclusion of *data citations in the reference list* to directly cite datasets that were re-used and obtained from public databases. Data citations in the article text are distinct from normal bibliographical citations and should directly link to the database records from which the data can be accessed. In the main text, data citations are formatted as follows: "Data ref: Smith et al, 2001" or "Data ref: NCBI Sequence Read Archive PRJNA342805, 2017". In the Reference list, data citations must be labeled with "[DATASET]". A data reference must provide the database name, accession number/identifiers and a resolvable link to the landing page from which the data can be accessed at the end of the reference. Further instructions are available at .

8) We replaced Supplementary Information with Expanded View (EV) Figures and Tables that are collapsible/expandable online. A maximum of 5 EV Figures can be typeset. EV Figures should be cited as 'Figure EV1, Figure EV2" etc... in the text and their respective legends should be included in the main text after the legends of regular figures.

- Additional Tables/Datasets should be labeled and referred to as Table EV1, Dataset EV1, etc. Legends have to be provided in a separate tab in case of .xls files. Alternatively, the legend can be supplied as a separate text file (README) and zipped together with the Table/Dataset file. See detailed instructions here:

9) The paper explained: EMBO Molecular Medicine articles are accompanied by a summary of the articles to emphasize the major findings in the paper and their medical implications for the non-specialist reader. Please provide a draft summary of your article highlighting

10) For more information: There is space at the end of each article to list relevant web links for further consultation by our readers. Could you identify some relevant ones and provide such information as well? Some examples are patient associations, relevant databases, OMIM/proteins/genes links, author's websites, etc...

11) Every published paper now includes a 'Synopsis' to further enhance discoverability. Synopses are displayed on the journal webpage and are freely accessible to all readers. They include a short stand first (maximum of 300 characters, including space) as well as 2-5 one-sentences bullet points that summarizes the paper. Please write the bullet points to summarize the key NEW findings. They should be designed to be complementary to the abstract - i.e. not repeat the same text. We encourage inclusion of key acronyms and quantitative information (maximum of 30 words / bullet point). Please use the passive voice. Please attach these in a separate file or send them by email, we will incorporate them accordingly.

Please also suggest a striking image or visual abstract to illustrate your article. If you do please provide a png file 550 px-wide x 400-px high.

12) As part of the EMBO Publications transparent editorial process initiative (see our Editorial at <http://embomolmed.embopress.org/content/2/9/329>), EMBO Molecular Medicine will publish online a Review Process File (RPF) to accompany accepted manuscripts.

In the event of acceptance, this file will be published in conjunction with your paper and will include the anonymous referee reports, your point-by-point response and all pertinent correspondence relating to the manuscript. Let us know whether you agree with the publication of the RPF and as here, if you want to remove or not any figures from it prior to publication.

I look forward to receiving your revised manuscript.

Yours sincerely,

Lise Roth

Lise Roth, PhD
Editor
EMBO Molecular Medicine

***** Reviewer's comments *****

Referee #1 (Comments on Novelty/Model System for Author):

Use of human samples with in vitro validation - very nice work from the authors.

Referee #1 (Remarks for Author):

This is a clinically relevant and well described study from Prof Jeyasekharan and colleagues.

Strengths include the interrogation of RAD51 in archival samples and in vitro, and cross validation in an independent cohort (exclusively platinum treated) with HRD scores available. Some minor limitations are noted below which could be addressed by the authors:

- 1) As a continuous score (using quartiles of expression) the utility as a predictive biomarker is limited. Were the authors able to compare scores between the two cohorts (BCC and SCOTROC) - were RAD51NES scores similar? Could a cut-off be defined? If scores were substantially different between the two cohorts (due to difference in antibody or storage of FFPE samples) could the authors discuss this limitation and what further work would be required to address this, as a point in the discussion?
- 2) In terms of the platinum sensitivity in vitro data - is this consistent with survival data, in that ovarian cell lines are often HRD and therefore RAD51 plays a lesser role (in keeping with the human data?). Sensitivity to platinum based on a 96 hour assay is less reliable than a 10-14 day clonogenic assay. I would suggest the authors perform this clonogenic assay in a similar manner to confirm no difference in sensitivity is observed.
- 3) The data on RAD51 and exclusion of cytotoxic T cells is very interesting, and reported in some detail. Inclusion of DAPI with an outline for tumour/stroma might aid interpretation of the main figure. While an absolute reduction in CD8+ cells is seen in the RAD51NES high quartile tumours, this is not the case for CD3 (alone) or FOXP3 cells, and it looks like there is an increase of intratumoral Tregs in RAD51NES high tumours? This is an interesting finding (if I am interpreting correctly) and

could be discussed by the authors.

4) I agree with the authors' point that layering of biomarkers leads to more interpretable and valuable data - however while the HRD score has a defined cut off, the lack of defined cut-off for RAD51 limits this applicability in the clinical setting.

5) The issue of taxol treatment should be discussed. While in the research setting, carboplatin alone is interesting, the majority of patients will receive combination taxol/platinum therapy. Presumably there are TILs data in support of the author's hypothesis for taxol/platinum treated ovarian cancer, suggesting that this assay may be more broadly applicable in taxol/platinum combination treated samples also.

6) This may be beyond the scope - are gene expression data (beyond HRD score) available for the RAD51NES high/ HR proficient tumours? Can this be explored for other potential targets in this poor prognosis platinum-resistant subgroup? A biomarker of resistance is limited in utility without alternative treatments being offered. A point for discussion perhaps.

7) What do the authors hypothesise is causing immune exclusion in RAD51NES-high tumours?

Referee #2 (Remarks for Author):

Hoppe et al. develop a quantitative assay for RAD51 imaging in ovarian cancer. They find an association of high RAD51 expression with progression free survival (PFS) and overall survival (OS) in a patient cohort treated with standard-of-care protocols. The authors reason that RAD51, being involved in the homologous repair (HR) pathway, might be associated with resistance to the platinum component of the treatment. To that end, they analyze the SCOTROC4 cohort which utilized carboplatin monotherapy. Here, again the authors find an association with poorer PFS and OS, specifically in HR proficient patients. The authors briefly investigate the mechanism by which RAD51 might confer resistance to platinum compounds, but find no effect of overexpression of RAD51 in vitro. Instead they find an enrichment of gene-expression signatures that might indicate a role of RAD51 in regulating immune function in tumors. An analysis of RAD51 high versus low tumors finds that the latter appear to exclude cytotoxic T-cells.

In principle, the discovery of a biomarker to predict therapy-resistance in (epithelial) ovarian cancer is of high interest, and the establishment and validation of a clinical-grade assay for RAD51 expression is well performed. However, there are a few points that the authors might want to address to increase the impact of the study. These mainly pertain to the clinical significance and the mechanism of action. Alternatively, the manuscript might be suitable to be published as a report without the detailed mechanistic analyses, however a further validation of the clinical significance would be beneficial to the potential impact:

Clinical significance:

One of the important findings appears that RAD51 can be used to stratify HR proficient according to platinum sensitivity. However, this finding is based on data of the SCOTROC4 cohort, using a carboplatin monotherapy protocol. Are the effects still visible in patients treated with a standard-of-care protocol? Can the authors stratify the BCC cohort or find a cohort for which HR scores are available, to confirm that a prognostic signal for RAD51 still exists independent of HR-status in those, clinically more relevant, patients?

Similarly, the immune infiltration data should also be performed for the BCC or a similar cohort. The authors focus on CD8 cells in their model for RAD51 action. However, in Supplemental Fig. 3B it is shown that also FOXP3+ Treg appear to be excluded from RAD51 high tumors. How would that fit into the model?

Mechanism of action of RAD51 overexpression:

The data on the immune infiltration and the mechanistic link to the gene-expression profiles in HRD51 overexpressing cells are quite weak and only correlative. The stronger level of immune-infiltration could also be a consequence of increased sensitivity of the tumors. The names of gene-sets in the enrichment analysis are not really informative. The authors should show details of the differentially regulated genes and provide the expression data and the analyses as supplementary information. Are the differentially regulated genes secreted/membrane factors that actually could have a functional role in modulating the immune response? Could the authors employ co-culture or other assays to demonstrate that RAD51 overexpression indeed has an impact on immune-function? The experiments in supplementary Fig. 1 are not really conclusive. The carboplatin-sensitivity should also be shown for an empty vector control. The RAD51 levels for Caov3 in the si control appear to be lower than in the siRAD51-CDS/FLAG-RAD51 cells. Yet there appears to be a difference in carboplatin sensitivity. How do the authors explain this? Could it be that the FLAG tag impairs the function of RAD51?

The absence of platinum resistance in vitro does not necessarily rule out that an immune-independent effect can be seen in vivo. Can the authors perform xenograft + treatment experiments?

Additional points:

Fig. 1B: A proper isotype control instead of no primary would be preferable

Fig. 3B: Label for X-axis is missing. The plot is not really informative, a table with all gene-sets and their enrichment scores as supplement would be more informative.

Page 7: I assume the authors mean '... 6 cycles of 3 weekly carboplatin ...' instead of paclitaxel

Page 28: The olaparib treatment is not shown in Fig. 2A

Supplementary Fig. 1a: Statistics are missing. Do the authors

Methods: Description of Western blot is missing

Referee #3:

This manuscript addresses a clear clinical issue of relapse on platinum based chemotherapy in EOC patients. If prognostic benefit is demonstrated in an appropriately controlled prospective clinical trial, the RAD51 scoring approach has the potential to be used for patient treatment decisions and could potentially be applied to other cancer types. It would also be interesting to understand if the RAD51 NES has benefit in the HRD positive sub-group at the time of progression on platinum or PARPi (ie demonstrate resistance is acquired). As mentioned, other studies have assessed digital pathology quantification of RAD51 but are limited to ex-vivo systems. The RAD51 NES therefore provides a novel approach for direct assessment of EOC patient material and the SCOTROC4 trial provides a robust way to measure the RAD51 scoring system in the context of platinum monotherapy.

The functional validation is currently weak and requires further work - Fig 3A should be repeated using a clonogenic assay, while the GSEA data generates interesting hypotheses but does not explain the discordance seen in-vitro and in patients for platinum resistance with RAD51 high expression. Due to this, there was no mechanistic work done to explain why RAD51 high EOC patients are more likely to relapse on platinum (e.g. evidence of enhanced adduct repair) and what an alternative therapy option for these patients might be.

The cytotoxic T-cell exclusion in RAD51 NES high patients finding is interesting but requires functional validation in-vivo (e.g. Assessing immune infiltrate in HGSOC RAD51 OE vs control

xenograft models and response to immune checkpoint inhibitors). The GSEA data from RAD51 OE cells in-vitro are also not demonstrated in patient samples, it would be important to understand whether RAD51 NES high tumours also display upregulation of immune modulatory genes and that this underpins the CD8+ exclusion.

As mentioned, this assay certainly has potential to be used for patient stratification. However, the difficulties in using an IF assay for clinical decision making are not discussed. Fluorescent intensity measurement is susceptible to tissue autofluorescence, scanning equipment used/fluorophore exposures and degradation of the fluorescent signal. These are significant challenges if an assay such as this were to be used for routine patient decision making across multiple sites.

* Figure 1 B - FFPE cell blocks lacking a positive control for RAD51 (e.g. treatment with MMC or irradiation)

* Fig 3A and Sup Fig 1B - clonogenic assays should be used, this will assess the long term replicative capacity of RAD51 OE cells following challenge with Carboplatin (RAD51 OE cells may be more resistant in terms of replicative capacity). Also why has the RAD51NES score not been used? Does rescue phenotype in sup Fig 1B correlate with increased RAD51NES? Would represent a functional validation of the RAD51NES in-vitro.

To

Dr. Lise Roth

Editor, EMBO Mol Med

Dear Dr. Roth,

Thank you for your email dated 21st Oct 2020, and for taking the time to speak this week. We are glad to hear that our study was of interest to the referees and the editorial team.

[...]

As suggested, below is a reply to the referee's comments, in the form of a prospective revision plan: to ensure that our revised version can be modified into a short report with appropriate changes.

We look forward to hearing if this will be acceptable for the revision.

Warm regards,

Anand

Prospective revision plan for the manuscript EMM-2020-13366 "Quantitative imaging of RAD51 expression as a marker of platinum resistance in ovarian cancer"

We are very grateful to the three referees for their careful analysis of our work, and for their constructive suggestions on how to improve the manuscript. Overall, we would like to proceed with the option of revising this into a short report focused on the quantitative IHC approach to measuring RAD51 in ovarian cancer, and its relevance to platinum resistance. Below is a point by point "revision plan", for consideration by the referees. We will be grateful for their advice on its adequacy, and will be happy to amend it as required. Our comments are in blue italics

Referee #1 (Comments on Novelty/Model System for Author):

Use of human samples with in vitro validation - very nice work from the authors.

Referee #1 (Remarks for Author):

This is a clinically relevant and well described study from Prof Jeyasekharan and colleagues. Strengths include the interrogation of RAD51 in archival samples and in vitro, and cross validation in an independent cohort (exclusively platinum treated) with HRD scores available.

We thank the referee for her/his kind comments, and are glad that the approach/ results are of interest

Some minor limitations are noted below which could be addressed by the authors:

1) As a continuous score (using quartiles of expression) the utility as a predictive biomarker is limited. Were the authors able to compare scores between the two cohorts (BCC and SCOTROC) - were RAD51NES scores similar? Could a cut-off be defined? If scores were substantially different between the two cohorts (due to difference in antibody or storage of FFPE samples) could the authors discuss this limitation and what further work would be required to address this, as a point in the discussion?

Yes. As predicted by the referee, the absolute scores between these two cohorts were quite different, likely due to sample preparation/ storage/ time since cutting of slides. While the aim of the current manuscript was to highlight the importance of RAD51 expression as a marker of platinum resistance, we agree that converting this into a robust assay for routine clinical use will require ideally a cut-off to dichotomize samples. This will require the setup of a protocol optimized for sample preparation, with suitable reference standard controls, and prospective validation. We thank the referee for highlighting this, and we will aim to address these points in the discussion as suggested.

2) In terms of the platinum sensitivity in vitro data - is this consistent with survival data, in that ovarian cell lines are often HRD and therefore RAD51 plays a lesser role (in keeping with the human data?). Sensitivity to platinum based on a 96 hour assay is less reliable than a 10-14 day clonogenic assay. I would suggest the authors perform this clonogenic assay in a similar manner to confirm no difference in sensitivity is observed.

The HRD status of these cell lines is indeed an interesting point. Among the 4 lines we used to create isogenic overexpression systems, COV362 is BRCA mutant while the rest are not. We do not have the HRD score for these cells, but the phenotypes we observed in-vitro thus far (MTT assay/ RNA sequencing) do not seem to be different between cell lines. We thank the referee for the suggestion of a clonogenic assay, and we will be completing this for the revised version.

3) The data on RAD51 and exclusion of cytotoxic T cells is very interesting, and reported in some detail. Inclusion of DAPI with an outline for tumour/stroma might aid interpretation of the main figure. While an absolute reduction in CD8+ cells is seen in the RAD51NES high quartile tumours, this is not the case for CD3 (alone) or FOXP3 cells, and it looks like there is an increase of intratumoral Tregs in RAD51NES high tumours? This is an interesting finding (if I am interpreting correctly) and could be discussed by the authors.

We thank the referee for the suggestion- and we will include the DAPI image with tumour/ stroma outlines. The exclusion from tumour regions is most statistically significant for the cytotoxic T-cells, but there is a trend towards exclusion of all CD3 cells and of CD3+FOXP3+ T-cells as well. We will discuss this in the light of new data we have on the gene expression profile of tumours with high RAD51.

4) I agree with the authors' point that layering of biomarkers leads to more interpretable and valuable data - however while the HRD score has a defined cut off, the lack of defined cut-off for RAD51 limits this applicability in the clinical setting.

Yes, we agree and we will highlight in the discussion the need to develop the RAD51 assay further into a binary readout (mentioned above in point 1). We will also discuss newer technologies (eg. digital spatial profiling) that may facilitate a cleaner "overall" readout.

5) The issue of taxol treatment should be discussed. While in the research setting, carboplatin alone is interesting, the majority of patients will receive combination taxol/platinum therapy. Presumably there are TILs data in support of the author's hypothesis for taxol/platinum treated ovarian cancer,

suggesting that this assay may be more broadly applicable in taxol/platinum combination treated samples also.

We apologize for not being clear in our writing. The immune infiltration studies were performed in the BCC cohort- of which the patients received standard Taxol-Platinum chemotherapy. We were unable to perform the immune infiltration studies on the SCOTROC4 cohort due to the limitations in number of TMA slides available. We will clarify this and highlight the differences between platinum only vs platinum-taxol cohorts to facilitate interpretation of the work.

6) This may be beyond the scope - are gene expression data (beyond HRD score) available for the RAD51NES high/ HR proficient tumours? Can this be explored for other potential targets in this poor prognosis platinum-resistant subgroup? A biomarker of resistance is limited in utility without alternative treatments being offered. A point for discussion perhaps.

7) What do the authors hypothesise is causing immune exclusion in RAD51NES-high tumours?

We thank the referee for this important suggestion. Gene expression studies are not available for these exact cohorts as they are formalin fixed samples in TMA format. However, we were able to perform a simple analysis by dividing tumours from the TCGA into high vs low quartiles for RAD51 mRNA, and then ask what other genes/ pathways were associated with high RAD51 mRNA expression. While RAD51 mRNA only correlates partly with RAD51 protein in tumours ($r=0.26$ in the TCGA for mRNA vs RPPA data), we did observe some interesting correlations:

1. *RAD51 high tumours showed enrichment/ loss of gene expression networks highlighted in the figure below:*

Many of these hallmark pathways (G2M, E2F, Spermatogenesis, Spindle etc) primarily represent shared regulation by the DREAM complex, which transcriptionally controls expression of several cell-cycle/ DNA repair genes. However, it was interesting to note enrichment of the IFN gamma and alpha signatures in this analysis.

2. *We subsequently analysed 4 independent cohorts of ovarian cancer with RNA data available, by dividing them into RAD51 high vs low (quartiles of mRNA) and then comparing the expression of a curated set of immune genes ($n\sim 800$). The results are presented below, both as individual genes and as key immune pathways that differentiate RAD51 high vs low tumours. Interestingly, there was a remarkable concordance between these 4 cohorts; with a consistent set of immune genes being associated with RAD51 mRNA levels. These immune genes are not regulated by the DREAM complex, so the association is unlikely to be co-transcriptional.*

We see two possible messages from this analysis:

- Firstly, genes mediating T-cell recruitment and antigen presentation were consistently high in the RAD51mRNA-high samples. Coupled with the data on T-cell exclusion that we observe by qIHC, we propose the possibility of an immune checkpoint that prevents T-cell infiltration into the tumour from the stroma, despite the ongoing chemotactic signal from CXCL9/10/11. It appears that these tumours are trying to get the T-cells in,

but cannot do so. In addition, our qIHC data suggests that this checkpoint is CD163 macrophage-independent.

- b. Secondly, there are certain immuno-oncology therapeutic targets (eg. CD47) that are consistently higher in the RAD51 mRNA-high group. We plan to touch upon this in the discussion as well*

We would like to add this mRNA analysis into the paper as a supplementary figure, while highlighting the limitations of the mRNA based correlative analysis. We hope to address the nature of this immune checkpoint in a future mechanistic study.

Referee #2 (Remarks for Author):

Hoppe et al. develop a quantitative assay for RAD51 imaging in ovarian cancer. They find an association of high RAD51 expression with progression free survival (PFS) and overall survival (OS) in a patient cohort treated with standard-of-care protocols. The authors reason that RAD51, being involved in the homologous repair (HR) pathway, might be associated with resistance to the platinum component of the treatment. To that end, they analyze the SCOTROC4 cohort which utilized carboplatin monotherapy. Here, again the authors find an association with poorer PFS and OS, specifically in HR proficient patients. The authors briefly investigate the mechanism by which RAD51 might confer resistance to platinum compounds, but find no effect of overexpression of RAD51 in vitro. Instead they find an enrichment of gene-expression signatures that might indicate a role of RAD51 in regulating immune function in tumors. An analysis of RAD51 high versus low tumors finds that the latter appear to exclude cytotoxic T-cells.

In principle, the discovery of a biomarker to predict therapy-resistance in (epithelial) ovarian cancer is of high interest, and the establishment and validation of a clinical-grade assay for RAD51 expression is well performed. However, there are a few points that the authors might want to address to increase the impact of the study. These mainly pertain to the clinical significance and the mechanism of action. Alternatively, the manuscript might be suitable to be published as a report without the detailed mechanistic analyses, however a further validation of the clinical significance would be beneficial to the potential impact:

We are grateful for the referee's interest in the work and kind comments on the establishment and validation of the assay. We are appreciative of the suggestion to convert the manuscript into a short report, allowing the possibility of more work towards a separate manuscript on the mechanistic link between RAD51 overexpression and immune exclusion.

Clinical significance:

One of the important findings appears that RAD51 can be used to stratify HR proficient according to platinum sensitivity. However, this finding is based on data of the SCOTROC4 cohort, using a carboplatin monotherapy protocol. Are the effects still visible in patients treated with a standard-of-care protocol? Can the authors stratify the BCC cohort or find a cohort for which HR scores are available, to confirm that a prognostic signal for RAD51 still exists independent of HR-status in those, clinically more relevant, patients?

The BCC cohort is a large and pathologically well-characterized set of cases provided by our collaborator Dr. David Huntsman, but unfortunately does not have HRD data available. The samples are already converted into TMA format, which is not compatible with HRD testing. Furthermore, HRD testing is still not routine at our institution, so it will also be difficult to generate a new cohort at NUH to confirm these findings.

Could we check with the referee if it will be feasible to address this as a point in the discussion? We will state that stratification of HR proficient cases by RAD51 is currently studied in a platinum only scenario, and that this will need to be prospectively validated in a platinum-taxol treated cohort. The need for a prospective cohort is also relevant to referee 1's points (1 and 4) about the RAD51 score needing to be optimized as a binary readout before being applied routinely along with HRD.

Similarly, the immune infiltration data should also be performed for the BCC or a similar cohort.

We apologize for the lack of clarity in our presentation/ writing. All the immune infiltration data was actually performed in the BCC cohort - which is platinum-taxol treated. We did not have adequate slides from the SCOTROC4 TMA to perform immune infiltration analyses, due to the paucity of available material from this clinical trial. The relevance of immune infiltration in the setting of platinum-taxol will be highlighted by mentioning the cohort name clearly in the text and the figures.

The authors focus on CD8 cells in their model for RAD51 action. However, in Supplemental Fig. 3B it is shown that also FOXP3+ Treg appear to be excluded from RAD51 high tumors. How would that fit into the model?

We thank the referee for highlighting this interesting point. As mentioned in our response above to referee 1 (points 6 and 7), our new RAD51 mRNA-correlated gene expression analysis seems to suggest that RAD51 high tumours have high levels of T-cell chemotactic factors CXCL9/10/11, but yet exhibit a generalized decrease in intra-tumoral T-cells. It is possible that the checkpoint responsible for the paucity of TILs is therefore independent of FOXP3 (and CD163 macrophages). We propose to add the gene expression data as a supplementary figure, highlighting the limitations of the RNA based correlative analysis, and use it to generate a possible model for future mechanistic studies.

Mechanism of action of RAD51 overexpression:

The data on the immune infiltration and the mechanistic link to the gene-expression profiles in HRD51 overexpressing cells are quite weak and only correlative. The stronger level of immune-infiltration could also be a consequence of increased sensitivity of the tumors. The names of gene-sets in the enrichment analysis are not really informative. The authors should show details of the differentially regulated genes and provide the expression data and the analyses as supplementary information. Are the differentially regulated genes secreted/membrane factors that actually could have a functional role in modulating the immune response? Could the authors employ co-culture or other assays to demonstrate that RAD51 overexpression indeed has an impact on immune-function?

The absence of platinum resistance in vitro does not necessarily rule out that an immune-independent effect can be seen in vivo. Can the authors perform xenograft + treatment experiments?

Yes, we agree with the referee that the data so far are correlative. Our new gene expression analysis showing that tumours with high RAD51 mRNA consistently have a specific pattern of immune gene expression (from bulk RNA data across 4 distinct cohorts) further strengthens the association, and highlights the need for further mechanistic work is required to understand the phenomenon.

The RNA seq in-vitro was done across 4 cell lines as biological replicates (vector control/ Flag-RAD51). Across cell lines, there was no standout mRNA that was consistently >2fold increased when RAD51 was overexpressed stably. There were small increases in several genes, and the only significance was noted when a pathway analyses was performed. We will include the actual fold change of all genes in a supplementary table as suggested. It is possible that the Flag-overexpression construct driven off a LTR promoter does not fully recapitulate the scenario of RAD51 overexpression that occurs in tumours and we will address this point in the writing of the revised version as well.

Overall, we would like to propose for the revised short report to focus on the clinical associations of survival and altered tumour immunity with RAD51 expression. We thank the referee for the suggestions on avenues to investigate the mechanism underlying these findings, and hope to collate co-culture experiments, humanized xenograft models and staining of early cancer/ dysplasia samples into a separate paper in the future (as these experiments are likely to take over a year to setup/ complete). For this current paper, we propose to complete clonogenic survival assays to check if there remains a subtle in-vitro fitness advantage in proliferation after chemotherapy exposure when RAD51 is high. We hope this will be acceptable.

The experiments in supplementary Fig. 1 are not really conclusive. The carboplatin-sensitivity should also be shown for an empty vector control. The RAD51 levels for Caov3 in the si control appear to be lower than in the siRAD51-CDS/FLAG-RAD51 cells. Yet there appears to be a difference in carboplatin sensitivity. How do the authors explain this? Could it be that the FLAG tag impairs the function of RAD51?

We apologize for the lack of clarity in explaining the experiment performed. The purpose of this experiment was to show that Flag-RAD51 was “functional”- able to rescue the increased platinum sensitivity when endogenous RAD51 was depleted. The differential depletion of endogenous RAD51 was performed by an siRNA to the 3' UTR, which did not affect the stable overexpressed version. The siRNA of RAD51 CDS was a control to deplete both overexpressed and endogenous RAD51.

In the western there are two sizes for RAD51. The smaller endogenous protein is seen in lane 1 for each cell line (transfected with the empty vector only). Lanes 2-4 have stable overexpression of Flag-RAD51. With the stable overexpression of Flag-RAD51 and siControl, a larger protein is noted, but the smaller endogenous protein is still present (lane 2). With siRNA to the CDS, both forms are depleted, to varying levels in different cell lines. With the siRNA to the 3'UTR, the larger Flag-RAD51 is unaffected but the endogenous form is depleted.

The cell viability experiments are shown only for the cells with stable overexpression of Flag-RAD51, to show that when both overexpressed and endogenous (siCDS) are depleted, the cells are more sensitive to platinum. When only the endogenous is depleted, they are not as sensitive, suggesting that the Flag-RAD51 is at least partially functional. The siControl in the viability experiments refers actually to siControl + pMSCV RAD51 in the western blots. We will address this by improving the labelling of the figure.

We also have data on the ability of Flag-RAD51 to form foci after platinum treatment (below), further demonstrating that this ectopically expressed RAD51 is functional, and can add this into the paper as a supplementary figure.

Additional points:

Fig. 1B: A proper isotype control instead of no primary would be preferable

Fig. 3B: Label for X-axis is missing. The plot is not really informative, a table with all gene-sets and their enrichment scores as supplement would be more informative.

Page 7: I assume the authors mean '... 6 cycles of 3 weekly carboplatin ...' instead of paclitaxel

Page 28: The olaparib treatment is not shown in Fig. 2A Supplementary Fig. 1a: Statistics are missing. Do the authors

Methods: Description of Western blot is missing

We thank the referee for highlighting these issues, and we will address all of them in the revised version.

Referee #3:

This manuscript addresses a clear clinical issue of relapse on platinum based chemotherapy in EOC patients. If prognostic benefit is demonstrated in an appropriately controlled prospective clinical trial, the RAD51 scoring approach has the potential to be used for patient treatment decisions and could potentially be applied to other cancer types. It would also be interesting to understand if the RAD51 NES has benefit in the HRD positive sub-group at the time of progression on platinum or PARPi (ie demonstrate resistance is acquired). As mentioned, other studies have assessed digital pathology quantification of RAD51 but are limited to ex-vivo systems. The RAD51 NES therefore provides a novel approach for direct assessment of EOC patient material and the SCOTROC4 trial provides a robust way to measure the RAD51 scoring system in the context of platinum monotherapy.

We thank the referee for their comments on the potential utility of measuring RAD51 nuclear expression in ovarian cancer. We agree that prospective studies will be the next step to validate the assay, and also define clear cut-offs for clinical use (as highlighted by referee 1 as well). Our current samples are all pre-treatment, so we are unable to directly address the question of secondary resistance in this manuscript. This is however an important point that we hope to study in future projects, and also evaluate in the context of PARP inhibitor resistance. We are attempting to setup collaborations with clinical trial groups to facilitate these studies, and we will incorporate this point in the revised discussion.

The functional validation is currently weak and requires further work - Fig 3A should be repeated using a clonogenic assay, while the GSEA data generates interesting hypotheses but does not explain the discordance seen in-vitro and in patients for platinum resistance with RAD51 high expression. Due to this, there was no mechanistic work done to explain why RAD51 high EOC patients are more likely to relapse on platinum (e.g. evidence of enhanced adduct repair) and what an alternative therapy option for these patients might be.

We thank the referee for the suggestion. These concerns have also been highlighted by referee 1 (comment 2). We will be performing the clonogenic assays to submit with the revised version of the paper, which is now being modified into a short report instead of a full manuscript.

The cytotoxic T-cell exclusion in RAD51 NES high patients finding is interesting but requires functional validation in-vivo (e.g. Assessing immune infiltrate in HGSOE RAD51 OE vs control xenograft models and response to immune checkpoint inhibitors).

We thank the referee for these suggestions. Overall, we propose to revise the manuscript to a short report to focus on the clinical associations of survival and altered tumour immunity with RAD51 expression. We do wish to investigate the mechanism underlying these findings, and if considered acceptable to the referees- we hope to collate co-culture experiments, humanized xenograft models and staining of early cancer/ dysplasia samples into a separate paper in the future (as these experiments are likely to take over a year to setup/ complete).

The GSEA data from RAD51 OE cells in-vitro are also not demonstrated in patient samples, it would be important to understand whether RAD51 NES high tumours also display upregulation of immune modulatory genes and that this underpins the CD8+ exclusion.

We thank the referee for this valuable suggestion, also echoed by referee 1. As our FFPE TMAs do not have gene expression data, we have attempted to study the gene expression of RAD51 high vs low tumours (albeit dividing RAD51 on the basis of mRNA and not NES). Our findings are summarized above in the response to referee 1 (comments 6 and 7).

As mentioned, this assay certainly has potential to be used for patient stratification. However, the difficulties in using an IF assay for clinical decision making are not discussed. Fluorescent intensity measurement is susceptible to tissue autofluorescence, scanning equipment used/fluorophore exposures and degradation of the fluorescent signal. These are significant challenges if an assay such as this were to be used for routine patient decision making across multiple sites.

We thank the referee for highlighting these important points. The spectral camera on the Vectra system allows us to define and unmix autofluorescence in our experiments, but this will indeed be an issue if non-spectral cameras are used. The OPAL-TSA staining method has proven to be relatively stable for a period of 3-6 months, but is sensitive to the duration between the time of cutting slides

and staining. We will address these in the discussion, along with possible design of a prospective study moving forward.

* Figure 1 B - FFPE cell blocks lacking a positive control for RAD51 (e.g. treatment with MMC or irradiation)

Our current FFPE cell-blocks focused on siRNA of RAD51 to mainly demonstrate that the antibody did indeed detect the RAD51 protein. Separately, we had also evaluated RAD51_{NES} in FFPE samples from ovarian cancer PDX's which were exposed to irradiation (IR) ex-vivo (after harvest). Consistent with western blot data, we observe an increase in RAD51_{NES} in samples subjected to DNA damage (RAD51_{NES} values stated in RAD51 unmixed monochrome images). These data can be included as a supplementary figure if acceptable.

* Fig 3A and Sup Fig 1B - clonogenic assays should be used, this will assess the long term replicative capacity of RAD51 OE cells following challenge with Carboplatin (RAD51 OE cells may be more resistant in terms of replicative capacity). Also why has the RAD51NES score not been used? Does rescue phenotype in sup Fig 1B correlate with increased RAD51NES? Would represent a functional validation of the RAD51NES in-vitro.

We thank the referee for the suggestions, and we will add clonogenic assays to the revised version of the paper. For the NES score, the values in the in-vitro cell blocks are significantly different from that of clinical tissue. We will nonetheless report the NES values for the knockdown samples, but do not think that these would represent a true validation of the score per se. We will highlight the need for prospective studies to refine the assay towards a clinically applicable cut-off, as also mentioned by referee 1.

12th Nov 20

Dear Anand,

I have now heard from referees #1 and #2, and they stated:

Referee #1:

I am very happy with proposed plan - I look forward to seeing the completed manuscript. I know these findings will be of great interest - the additional mRNA analysis performed provides valuable data.

Referee #2:

In my opinion the authors provide a reasonable way forward to revise the manuscript into a short report. Leaving the mechanistic studies for a future manuscript definitely makes sense given the number and challenge of the necessary experiments, especially under the current conditions.

The clarifications, changes and potential additions to the manuscript, as suggested by the authors, would address my concerns.

Both referees approve your revision plan, and we therefore look forward receiving your manuscript revised along these lines.

With my best wishes,

Lise

Lise Roth, PhD

Editor

EMBO Molecular Medicine

l.roth@embomolmed.org

Revised manuscript EMM-2020-13366 “Quantitative imaging of RAD51 expression as a marker of platinum resistance in ovarian cancer”

We are very grateful to the three referees for their careful analysis of our work, and for their constructive suggestions on how to improve the manuscript. As indicated in our prospective revision plan, we have revised our manuscript into short report that focuses on the quantitative IHC approach to measuring RAD51 in ovarian cancer, and its relevance to platinum resistance. Below is a point-by-point response to each of the referee’s queries and detailing how we have modified the manuscript accordingly.

Our comments are in blue italics, while relevant sections from the paper are in black italics

Referee #1 (Comments on Novelty/Model System for Author):

Use of human samples with in vitro validation - very nice work from the authors.

Referee #1 (Remarks for Author):

This is a clinically relevant and well described study from Prof Jeyasekharan and colleagues. Strengths include the interrogation of RAD51 in archival samples and in vitro, and cross validation in an independent cohort (exclusively platinum treated) with HRD scores available.

We thank the referee for her/his kind comments, and are glad that the approach/ results are of interest

Some minor limitations are noted below which could be addressed by the authors:

1) As a continuous score (using quartiles of expression) the utility as a predictive biomarker is limited. Were the authors able to compare scores between the two cohorts (BCC and SCOTROC) - were RAD51NES scores similar? Could a cut-off be defined? If scores were substantially different between the two cohorts (due to difference in antibody or storage of FFPE samples) could the authors discuss this limitation and what further work would be required to address this, as a point in the discussion?

Yes. As predicted by the referee, the absolute scores between these two cohorts were quite different, likely due to sample preparation/ storage/ time since cutting of slides. While the aim of the current manuscript was to highlight the importance of RAD51 expression as a marker of platinum resistance, we agree that converting this into a robust assay for routine clinical use will require ideally a cut-off

to dichotomize samples. This will require the setup of a protocol optimized for sample preparation, with suitable reference standard controls, and prospective validation. We thank the referee for highlighting this and this point has been highlighted in the revision of our manuscript as follows:

'A limitation of our study is the lack of a RAD51_{NES} "cut-off" which unequivocally denotes resistance to platinum therapy. Defining a cut-off requires a prospective study with standardized protocols optimized for sample preparation and suitable reference standard controls. Furthermore, our protocol necessitates the use of a spectral camera to define and unmix autofluorescence- a common technical drawback of fluorescent IHC. Having identified RAD51 as a key determinant of platinum resistance using the multispectral method, a future comparison of methods for quantitative measurement (e.g., digital spatial profiling/ non-spectral qIHC) should evaluate which would be best for a robust determination of a cut-off for clinical use. Ideally this would be done in prospective cohorts from clinical trials of platinum/taxol and PARPi treatment in EOC, with availability of HRD scores to validate the relevance of the High-RAD51/HRD negative "subgroup".'

2) In terms of the platinum sensitivity in vitro data - is this consistent with survival data, in that ovarian cell lines are often HRD and therefore RAD51 plays a lesser role (in keeping with the human data?). Sensitivity to platinum based on a 96 hour assay is less reliable than a 10-14 day clonogenic assay. I would suggest the authors perform this clonogenic assay in a similar manner to confirm no difference in sensitivity is observed.

The HRD status of these cell lines is indeed an interesting point. Among the 4 lines we used to create isogenic overexpression systems, COV362 is BRCA mutant while the rest are not. We do not have the HRD score for these cells, but the phenotypes we observed in-vitro thus far (MTT assay/ RNA sequencing) do not seem to be different between cell lines. We thank the referee for the suggestion of a clonogenic assay. We performed this assay in two cell lines (Caov-3 and TYNK-u). Caov-3 cells formed clear colonies, and as demonstrated in what is now Figure 3B in the revised manuscript (below) there was no significant difference between the wild type and RAD51 overexpressing cells upon carboplatin treatment (n=4 independent experiments, growth of 2 weeks each). This consistent with our short-term cell viability assays. Unfortunately, the TYNK-u cells do not form clear colonies, and thus were not included in the manuscript- but visually we do not see a clear difference in growth between WT and RAD51 overexpressing TyKNU cells- data included below for your reference.

3) The data on RAD51 and exclusion of cytotoxic T cells is very interesting, and reported in some detail. Inclusion of DAPI with an outline for tumour/stroma might aid interpretation of the main figure. While an absolute reduction in CD8+ cells is seen in the RAD51NES high quartile tumours, this is not the case for CD3 (alone) or FOXP3 cells, and it looks like there is an increase of intratumoral Tregs in RAD51NES high tumours? This is an interesting finding (if I am interpreting correctly) and could be discussed by the authors.

We thank the referee for the suggestion, we have now amended this figure to include DAPI and outlines to denote both tumour and stroma (now Figure 3G of the revised manuscript- below).

The exclusion from tumour regions is most statistically significant and obvious for the cytotoxic T-cells, but there is a trend towards exclusion of all CD3 cells and of CD3+FOXP3+ T-cells as well. This same trend is not seen with macrophages, where there is little difference between RAD51 high and low tumours, or at the very least, a trend towards more macrophages infiltrating the tumour. It seems that the immune exclusion phenotype is for T-cell populations in general, although most critical for Cyt-Ts. We also have new gene expression profile data suggesting that this occurs despite an increase in chemokines that typically promote T-cell recruitment. We have incorporated this data and discussion into the revised manuscript (see response to points 6 and 7).

4) I agree with the authors' point that layering of biomarkers leads to more interpretable and valuable data - however while the HRD score has a defined cut off, the lack of defined cut-off for RAD51 limits this applicability in the clinical setting.

Yes, we agree and we have highlighted in the discussion the need and possible approaches to further develop the RAD51 assay further into a clinically applicable binary readout.

5) The issue of taxol treatment should be discussed. While in the research setting, carboplatin alone is interesting, the majority of patients will receive combination taxol/platinum therapy. Presumably there are TILs data in support of the author's hypothesis for taxol/platinum treated ovarian cancer, suggesting that this assay may be more broadly applicable in taxol/platinum combination treated samples also.

We apologize for not being clear in our writing. The immune infiltration studies were performed in the BCC cohort- of which the patients received standard taxol-platinum chemotherapy. We have now clarified this in the figure and figure legend.

6) This may be beyond the scope - are gene expression data (beyond HRD score) available for the RAD51NES high/ HR proficient tumours? Can this be explored for other potential targets in this poor prognosis platinum-resistant subgroup? A biomarker of resistance is limited in utility without alternative treatments being offered. A point for discussion perhaps.

7) What do the authors hypothesise is causing immune exclusion in RAD51NES-high tumours?

We thank the referee for this important suggestion. Gene expression studies are not available for these exact cohorts as they are formalin fixed samples in TMA format. However, we were able to perform a simple analysis by dividing tumours from the TCGA into high vs low quartiles for RAD51 mRNA, and then ask what other genes/ pathways were associated with high RAD51 mRNA expression. We observed some interesting correlations:

1. RAD51 high tumours showed enrichment/ loss of gene expression networks highlighted in the figure below (Figure 3D for the revised manuscript):

Many of these hallmark pathways (G2M, E2F, Spindle etc) primarily represent shared regulation by the DREAM complex, which transcriptionally controls expression of several cell-cycle/ DNA repair genes. However, it was interesting to note enrichment of the IFN gamma and alpha signatures in this analysis. See also Table EV3 in the revised manuscript.

2. We subsequently analysed 4 independent cohorts of ovarian cancer with RNA data available, by dividing them into RAD51 high vs low (quartiles of mRNA) and then comparing the expression of a curated set of immune genes ($n \sim 800$). The results are presented below as individual genes that differentiate RAD51 high vs low tumours (incorporated into the revised manuscript as Figures 3E, F and Figure EV4). Interestingly, there was a remarkable concordance between these 4 cohorts; with a consistent set of immune genes being associated with RAD51 mRNA levels. These immune genes are not regulated by the DREAM complex, so the association is unlikely to be co-transcriptional.

Our interpretation of the above data in conjunction with the qIHC immune exclusion data has been included in revised manuscript as follows (key sections highlighted in yellow):

Corroborating this, in clinical samples of the TCGA HGSOc cohort, among the pathways highly associated with RAD51 were interferon responses (Fig 3D, Table EV3). We then interrogated a

curated set of immune-related genes with RAD51 in four distinct mRNA cohorts of EOC (TCGA, AOCs, MGH, Duke) and identified a remarkably consistent enrichment of specific immune genes in RAD51-High tumours (Fig 3E and 3F; Fig EV4; Table EV4). These included consistent upregulation of genes regulating antigen presentation (TFRC, PSMB7/9, TAP1/2) and chemokines (CXCL10, CCL8), but also of certain negative immune-checkpoints such as CD47. We therefore aimed to further define the immune microenvironment of high-RAD51 cancers at single-cell resolution using multispectral qIHC. We focused on T-cells and immunosuppressive macrophages due to their known prognostic role in EOC, using the immune markers CD3, CD8, FOXP3 and CD163 along with cytokeratin as a tumour mask to separate stromal and tumour compartments (Fig EV5A). Using the platinum-taxol treated BCC cohort, we observed a significant exclusion of CD3+/CD8+ cytotoxic T-cells from the tumour regions in RAD51-High cancers (Fig.3G-H). A similar but less prominent effect was noted with the total T-cell population (CD3+ only) and CD3+/FOXP3+ T regulatory cells, but not with CD163+ macrophages (Fig EV5BA). Our results mirror prior work in lung cancer, where low RAD51 was associated with increased TILs (Gachechiladze et al, 2020). The cytotoxic T-cell exclusion phenotype in our RAD51-High cases was primarily noted in BRCA wild-type tumours (Fig EV7C), in keeping with the prognostic significance of RAD51 in HRD negative patients, who are typically BRCA WT (Fig 2E). We speculate that high-RAD51 promotes an as yet unknown immune checkpoint that prevents T-cell infiltration (predominantly cytotoxic T-cells, but also other T-cell subsets) into the tumour from the stroma. The elevated CXCL chemotactic signals and expression of antigen presenting genes may represent an “ineffective” compensation to this negative checkpoint, ultimately resulting in evasion of immune surveillance and poor survival. These results also point to possible therapeutic approaches for RAD51-High tumours e.g., the addition of CTLA4 inhibitors to promote T-cell recruitment. The consistent correlation of CD47 expression with RAD51 is also interesting, and further work will be required to understand its biological significance in this setting along with the potential applicability of anti-CD47 monoclonal antibodies. Finally, given the proven clinical utility of anti-angiogenic drugs (e.g. Bevacizumab) in EOC, it will be interesting to evaluate whether their role in remodelling tumour vasculature to facilitate T-cell migration (Dickson et al, 2007; Wallin et al, 2016) can specifically overcome the platinum resistance seen in high-RAD51 EOC.'

We hope to address the nature of this immune checkpoint in a future mechanistic study.

Referee #2 (Remarks for Author):

Hoppe et al. develop a quantitative assay for RAD51 imaging in ovarian cancer. They find an association of high RAD51 expression with progression free survival (PFS) and overall survival (OS) in a patient cohort treated with standard-of-care protocols. The authors reason that RAD51, being involved in the homologous repair (HR) pathway, might be associated with resistance to the platinum component of the treatment. To that end, they analyze the SCOTROC4 cohort which utilized carboplatin monotherapy. Here, again the authors find an association with poorer PFS and OS, specifically in HR proficient patients. The authors briefly investigate the mechanism by which RAD51 might confer resistance to platinum compounds, but find no effect of overexpression of RAD51 in vitro. Instead they find an enrichment of gene-expression signatures that might indicate a role of RAD51 in regulating immune function in tumors. An analysis of RAD51 high versus low tumors finds that the latter appear to exclude cytotoxic T-cells.

In principle, the discovery of a biomarker to predict therapy-resistance in (epithelial) ovarian cancer is of high interest, and the establishment and validation of a clinical-grade assay for RAD51 expression is well performed. However, there are a few points that the authors might want to address to increase the impact of the study. These mainly pertain to the clinical significance and the mechanism of action. Alternatively, the manuscript might be suitable to be published as a report without the detailed mechanistic analyses, however a further validation of the clinical significance would be beneficial to the potential impact:

We are grateful for the referee's interest in the work and kind comments on the establishment and validation of the assay. We are appreciative of the suggestion to convert the manuscript into a short report, allowing the possibility of more work towards a separate manuscript on the mechanistic link between RAD51 overexpression and immune exclusion. We have followed this advice accordingly.

Clinical significance:

1) One of the important findings appears that RAD51 can be used to stratify HR proficient according to platinum sensitivity. However, this finding is based on data of the SCOTROC4 cohort, using a carboplatin monotherapy protocol. Are the effects still visible in patients treated with a standard-of-care protocol? Can the authors stratify the BCC cohort or find a cohort for which HR scores are available, to confirm that a prognostic signal for RAD51 still exists independent of HR-status in those, clinically more relevant, patients?

The BCC cohort is a large and pathologically well-characterized set of cases provided by our collaborator Dr. David Huntsman, but unfortunately does not have HRD data available. The samples are already converted into TMA format, which is not compatible with HRD testing. Furthermore, HRD testing is still not routine in clinical practice, so we are unable to generate a new cohort at NUH to confirm these findings. We have acknowledged this limitation in the revised manuscript:

'Ideally this would be done in prospective cohorts from clinical trials of platinum/taxol and PARPi treatment in EOC, with availability of HRD scores to validate the relevance of the High-RAD51/HRD negative "subgroup".'

The need for a prospective cohort is also relevant to referee 1's points (1 and 4) about the RAD51 score needing to be optimized as a binary readout before being applied routinely along with HRD.

2) Similarly, the immune infiltration data should also be performed for the BCC or a similar cohort.

We apologize for the lack of clarity in our presentation/ writing. All the immune infiltration data was actually performed in the BCC cohort - which is taxol-platinum treated. We did not have adequate slides from the SCOTROC4 TMA to perform immune infiltration analyses, due to the paucity of available material from this clinical trial. In the figures and figure legends we have clearly indicated that the immune infiltration data has been obtained from the taxol-platinum treated BCC cohort.

3) The authors focus on CD8 cells in their model for RAD51 action. However, in Supplemental Fig. 3B it is shown that also FOXP3+ Treg appear to be excluded from RAD51 high tumors. How would that fit into the model?

We thank the referee for highlighting this interesting point. As mentioned in our response above to referee 1 (points 6 and 7), our new RAD51 mRNA-correlated gene expression analysis seems to suggest that RAD51 high tumours have high levels of T-cell chemotactic factors CXCL10 and CLL8, but yet exhibit a generalized decrease in intra-tumoral T-cells. It is possible that the checkpoint responsible for the paucity of TILs is therefore independent of FOXP3 (and CD163 macrophages) but appears to affect T-cells specifically. We have added this gene expression data to the revised manuscript as Figures 3D, 3E, 3F and EV4; and Tables EV3 and EV4 (see response to referee 1, point 6 and 7). We will use this data to generate a possible model for future mechanistic studies.

Mechanism of action of RAD51 overexpression:

4) The data on the immune infiltration and the mechanistic link to the gene-expression profiles in HRD51 overexpressing cells are quite weak and only correlative. The stronger level of immune-infiltration could also be a consequence of increased sensitivity of the tumors. The names of gene-sets in the enrichment analysis are not really informative. The authors should show details of the differentially regulated genes and provide the expression data and the analyses as supplementary information. Are the differentially regulated genes secreted/membrane factors that actually could have a functional role in modulating the immune response? Could the authors employ co-culture or other assays to demonstrate that RAD51 overexpression indeed has an impact on immune-function?

The absence of platinum resistance in vitro does not necessarily rule out that an immune-independent effect can be seen in vivo. Can the authors perform xenograft + treatment experiments?

Yes, we agree with the referee that the data so far are correlative. Our new gene expression analysis showing that tumours with high RAD51 mRNA consistently have a specific pattern of immune gene expression (from bulk RNA data across 4 distinct cohorts) further strengthens the association, and highlights the need for further mechanistic work is required to understand the phenomenon (see Figures 3D, 3E, 3F and EV4; and Tables EV3 and EV4 of the revised manuscript).

The RNA seq in-vitro was done across 4 cell lines as biological replicates (vector control/ Flag-RAD51). Across cell lines, there was no standout mRNA that was consistently >2fold increased when RAD51 was overexpressed stably. There were small increases in several genes, and the only significance was noted when a pathway analyses was performed. In the revised manuscript we have now included the full GSEA analysis (Table EV1) and actual fold change of all individual genes in the analysis (Table EV2).

Overall, we have revised this manuscript into a short report that focuses on the clinical associations of survival and altered tumour immunity with RAD51 expression. We thank the referee for the suggestions on avenues to investigate the mechanism underlying these findings, and hope to collate co-culture experiments, humanized xenograft models and staining of early cancer/ dysplasia samples into a separate paper in the future. For this current paper, we have completed clonogenic survival

assays to check if there remains a subtle in-vitro fitness advantage in proliferation after chemotherapy exposure when RAD51 is high. As from Figure 3B of the revised manuscript, there is no obvious difference in survival between wild type and RAD51 overexpressing cells after carboplatin treatment (for the figures see response to referee 1, point 2).

5) The experiments in supplementary Fig. 1 are not really conclusive. The carboplatin-sensitivity should also be shown for an empty vector control. The RAD51 levels for Caov3 in the si control appear to be lower than in the siRAD51-CDS/FLAG-RAD51 cells. Yet there appears to be a difference in carboplatin sensitivity. How do the authors explain this? Could it be that the FLAG tag impairs the function of RAD51?

We apologize for the lack of clarity in explaining the experiment performed. The purpose of this experiment was to show that Flag-RAD51 was “functional”- able to rescue the increased platinum sensitivity when endogenous RAD51 was depleted. The differential depletion of endogenous RAD51 was performed by an siRNA to the 3’ UTR, which did not affect the stable overexpressed version. The siRNA of RAD51 CDS was a control to deplete both overexpressed and endogenous RAD51.

In the Western blot there are two sizes for RAD51. The smaller endogenous protein is seen in lane 1 for each cell line (transfected with the empty vector only). Lanes 2-4 have stable overexpression of Flag-RAD51. With the stable overexpression of Flag-RAD51 and siControl, a larger protein is noted, but the smaller endogenous protein is still present (lane 2). With siRNA to the CDS, both forms are depleted, to varying levels in different cell lines. With the siRNA to the 3’UTR, the larger Flag-RAD51 is unaffected but the endogenous form is depleted.

The cell viability experiments are shown only for the cells with stable overexpression of Flag-RAD51, to show that when both overexpressed and endogenous (siCDS) are depleted, the cells are more sensitive to platinum. When only the endogenous is depleted, they are not as sensitive, suggesting that the Flag-RAD51 is at least partially functional. The siControl in the viability experiments refers actually to siControl + pMSCV RAD51 in the western blots. We have improved the labelling of what is now Figure EV3B of the revised manuscript (below) to indicate that the cell lines being used for this experiment are all RAD51 overexpressing - pMSCV Flag-RAD51.

We also have data on the ability of Flag-RAD51 to form foci after platinum treatment (below), further demonstrating that this ectopically expressed RAD51 is functional, and have added this into the paper as Figure EV3C.

6) Additional points:

We thank the referee for highlighting the following issues; all have been addressed in the revised manuscript as described below:

Fig. 1B: A proper isotype control instead of no primary would be preferable

An isotype control has been incorporated into what is now Figure EV1B of the revised manuscript (below):

Fig. 3B: Label for X-axis is missing. The plot is not really informative, a table with all gene-sets and their enrichment scores as supplement would be more informative.

We have improved this figure, which is now Figure 3C of the revised manuscript, to include a label for the x-axis as well as the specific enrichment scores (ES) and false discovery rates (FDR) of the top rank gene sets (below):

We have also included the full GSEA analysis in Table EV1.

Page 7: I assume the authors mean '... 6 cycles of 3 weekly carboplatin ...' instead of paclitaxel
 Page 28: The olaparib treatment is not shown in Fig. 2A
 Supplementary Fig. 1a: Statistics are missing. Do the authors

This sentence no longer appears in the text

Methods: Description of Western blot is missing

This has now been included

Referee #3:

1) This manuscript addresses a clear clinical issue of relapse on platinum based chemotherapy in EOC patients. If prognostic benefit is demonstrated in an appropriately controlled prospective clinical trial, the RAD51 scoring approach has the potential to be used for patient treatment decisions and could potentially be applied to other cancer types. It would also be interesting to understand if the RAD51 NES has benefit in the HRD positive sub-group at the time of progression on platinum or PARPi (ie demonstrate resistance is acquired). As mentioned, other studies have assessed digital pathology quantification of RAD51 but are limited to ex-vivo systems. The RAD51 NES therefore provides a novel approach for direct assessment of EOC patient material and the SCOTROC4 trial provides a robust way to measure the RAD51 scoring system in the context of platinum monotherapy.

We thank the referee for their kind comments on the potential utility of measuring RAD51 nuclear expression in ovarian cancer. We agree that prospective studies will be the next step to validate the assay, and also define clear cut-offs for clinical use (as highlighted by referee 1 as well). Our current samples are all pre-treatment, so we are unable to directly address the question of secondary resistance in this manuscript. This is however an important point that we hope to study in future projects, and also evaluate in the context of PARP inhibitor resistance. We are attempting to setup collaborations with clinical trial groups to facilitate these future studies.

2) The functional validation is currently weak and requires further work - Fig 3A should be repeated using a clonogenic assay, while the GSEA data generates interesting hypotheses but does not explain the discordance seen in-vitro and in patients for platinum resistance with RAD51 high expression. Due to this, there was no mechanistic work done to explain why RAD51 high EOC patients are more likely to relapse on platinum (e.g. evidence of enhanced adduct repair) and what an alternative therapy option for these patients might be.

We thank the referee for the suggestion. These concerns have also been highlighted by referee 1 (point 2) and in response we have performed clonogenic assays (now Figure 3B). As with the short term viability assays, there is no obvious difference in survival after carboplatin treatment between wild type and RAD51 overexpressing cells (for the figures see response to referee 1, point 2).

With respect the mechanistic aspect of the paper, we agree that more work needs to be done to elucidate a full mechanism. We have therefore revised this paper into a short report focusing on the association of RAD51_{NES} and survival in EOC but also briefly discussing our immune related data as a prelude to a future mechanistic paper. We have also touched upon possible therapeutic options for these cases to overcome the immune exclusion phenotype.

3) The cytotoxic T-cell exclusion in RAD51 NES high patients finding is interesting but requires functional validation in-vivo (e.g. Assessing immune infiltrate in HGSOC RAD51 OE vs control xenograft models and response to immune checkpoint inhibitors).

We thank the referee for these valuable suggestions and directions for the work. In this current short report, we have attempted to focus on the clinical associations of survival and altered tumour immunity with RAD51 expression. We plan to, in future work, investigate the mechanism underlying these findings using co-culture experiments, humanized xenograft models and staining of early cancer/ dysplasia samples and report them in a separate paper.

4) The GSEA data from RAD51 OE cells in-vitro are also not demonstrated in patient samples, it would be important to understand whether RAD51 NES high tumours also display upregulation of immune modulatory genes and that this underpins the CD8+ exclusion.

We thank the referee for this valuable suggestion, also echoed by referee 1. As our FFPE TMAs do not have gene expression data, we have attempted to study the gene expression of RAD51 high vs low tumours (albeit dividing RAD51 on the basis of mRNA and not NES). Our findings are summarized above in the response to referee 1 (comments 6 and 7) and we have included this data in the paper in Figures 3D, 3E, 3F and EV4; and Tables EV3 and EV4.

5) As mentioned, this assay certainly has potential to be used for patient stratification. However, the difficulties in using an IF assay for clinical decision making are not discussed. Fluorescent intensity measurement is susceptible to tissue autofluorescence, scanning equipment used/fluorophore exposures and degradation of the fluorescent signal. These are significant challenges if an assay such as this were to be used for routine patient decision making across multiple sites.

We thank the referee for highlighting these important points. Below is our revised paragraph in the manuscript addressing these issues for clinical implementation:

'A limitation of our study is the lack of a RAD51_{NES} "cut-off" which unequivocally denotes resistance to platinum therapy. Defining a cut-off requires a prospective study with standardized protocols optimized for sample preparation and suitable reference standard controls. Furthermore, our protocol necessitates the use of a spectral camera to define and unmix autofluorescence- a common technical drawback of fluorescent IHC. Having identified RAD51 as a key determinant of platinum resistance using the multispectral method, a future comparison of methods for quantitative measurement (e.g., digital spatial profiling/ non-spectral qIHC) should evaluate which would be best for a robust determination of a cut-off for clinical use. Ideally this would be done in prospective cohorts from clinical trials of platinum/taxol and PARPi treatment in EOC, with availability of HRD scores to validate the relevance of the High-RAD51/HRD negative "subgroup".

6) Figure 1 B - FFPE cell blocks lacking a positive control for RAD51 (e.g. treatment with MMC or irradiation)

We thank the referee for this suggestion. Our current FFPE cell-blocks focused on siRNA of RAD51 to mainly demonstrate that the antibody did indeed detect the RAD51 protein. Separately, we had also evaluated RAD51_{NES} in FFPE samples from ovarian cancer PDX's which were exposed to irradiation (IR) ex-vivo (after harvest). Consistent with Western blot data, we observe an increase in RAD51_{NES} in samples subjected to DNA damage. This data is now included in the revised manuscript as Figure EV1C (below):

7) Fig 3A and Sup Fig 1B - clonogenic assays should be used, this will assess the long term replicative capacity of RAD51 OE cells following challenge with Carboplatin (RAD51 OE cells may be more resistant in terms of replicative capacity). Also why has the RAD51NES score not been used? Does rescue phenotype in sup Fig 1B correlate with increased RAD51NES? Would represent a functional validation of the RAD51NES in-vitro.

We thank the referee for the suggestions, we have added the clonogenic assays into the manuscript as described above in point 2 (and in response to referee 1, point 2). For the NES score, the values in the in-vitro cell blocks are significantly different from that of clinical tissue and thus we do not believe that they will represent a true validation of the score per se. In the revised manuscript we have also acknowledged the need to establish a reliable RAD51_{NES} cut-off value above which unequivocally denotes platinum resistance (also see response to referee 1, point 1).

21st Jan 2021

Dear Dr. Jeyasekharan,

Thank you for the submission of your revised manuscript to EMBO Molecular Medicine. We have now received the enclosed reports from the two referees who re-reviewed your manuscript. As you will see, they are both supportive of publication, and I am therefore pleased to inform you that we will be able to accept your manuscript, once the following editorial points will be addressed:

1) Main manuscript text:

- Please answer/correct the changes suggested by our data editors in the main manuscript file (in track changes mode). This file will be sent to you in the next couple of days. Please use this file for any further modification.

- Please remove the highlighted text.

- We can accommodate a maximum of 5 keywords, please adjust accordingly.

- Please move the Material and Methods section up, before Figure Legends. EV figure legends should directly follow the main figure legends.

- Material and Methods:

- o Patients samples: Please identify the committee(s) approving the study protocol. Please include the full statement that informed consent was obtained from all subjects and the experiments conformed to the principles set out in the WMA Declaration of Helsinki and the Department of Health and Human Services Belmont Report.

- o Cells: please indicate the origin of the cells, and whether they were authenticated and tested for mycoplasma contamination.

- o Mice: Please provide the housing and husbandry conditions. Please indicate the gender and origin of the mice used in your experiments.

- Data Availability Section: Please note that your RNAseq data have to be deposited in a public repository before publication of your manuscript. (see

<https://www.embopress.org/page/journal/17574684/authorguide#dataavailability>).

2) Figures:

- Please add (and define) a scale bar in Fig EV5C.

- Please correct the typos in the legend of Fig. EV2F (indicated Fig. EV2B) and in the main text, Fig. EV7C (should be Fig. EV5C?).

3) Appendix: Please include a table of content and update the nomenclature to "Appendix Table S1" etc. Please make sure that the Appendix tables are correctly referenced to in the main text.

4) Checklist:

- Section E/12: please include the full statement that informed consent was obtained from all subjects and the experiments conformed to the principles set out in the WMA Declaration of Helsinki and the Department of Health and Human Services Belmont Report.

- Please fill in the section D/Animal Models

- Please fill in section F/18 and F/19 on data availability. RNAseq data produced in this study have to be deposited in a public repository.

5) Please note that all corresponding authors are required to supply an ORCID ID for their name

upon submission of a revised manuscript. An ORCID identifier is missing for Dr. David Shao Peng Tan.

6) As you did not list any link, you may remove the section "For more information".

7) Thank you for providing a nice synopsis picture. When resized to 550 px-wide, the text of the legend (stromal cells, cancer cells, ..) becomes difficult to read. Could you please increase the font size?

8) As part of the EMBO Publications transparent editorial process initiative (see our Editorial at <http://embomolmed.embopress.org/content/2/9/329>), EMBO Molecular Medicine will publish online a Review Process File (RPF) to accompany accepted manuscripts.

In the event of acceptance, this file will be published in conjunction with your paper and will include the anonymous referee reports, your point-by-point response and all pertinent correspondence relating to the manuscript. Let us know whether you agree with the publication of the RPF and as here, **IF YOU WANT TO REMOVE OR NOT ANY FIGURES** from it prior to publication.

I look forward to receiving your revised manuscript.

Yours sincerely,

Lise Roth

Lise Roth, PhD
Editor
EMBO Molecular Medicine

***** Reviewer's comments *****

Referee #1 (Remarks for Author):

I enjoyed reading this revised manuscript - the authors should be complimented on the high standard of their work and discussion of results.

Referee #2 (Remarks for Author):

The manuscript, as revised into a short report, significantly strengthens the focus on the important finding- The authors also included additional data and clarifications in response to the reviewer's comments. All of my concerns have been sufficiently addressed and I now recommend this manuscript for publication.

The authors performed the requested editorial changes.

9th Feb 2021

Dear Anand,

Thank you for sending the revised files. I looked at everything and all is fine. I am thus very pleased to accept your manuscript for publication in EMBO Molecular Medicine!

As previously mentioned, your manuscript will be sent to our publisher once the GEO dataset will be public, therefore please kindly notify our editorial office as soon as this is done.

As part of the EMBO Publications transparent editorial process initiative (see our Editorial at <http://embomolmed.embopress.org/content/2/9/329>), EMBO Molecular Medicine will publish online a Review Process File (RPF) to accompany accepted manuscripts.

This file will be published in conjunction with your paper and will include the anonymous referee reports, your point-by-point response and all pertinent correspondence relating to the manuscript. Let us know whether you agree with the publication of the RPF and as here, if you want to remove or not any figures from it prior to publication.

Congratulations on a nice study!

With my best wishes,

Lise

Lise Roth, Ph.D
Editor
EMBO Molecular Medicine

Follow us on Twitter @EmboMolMed
Sign up for eTOCs at embopress.org/alertsfeeds

Corresponding Author Name: Anand Jeyasekharan

Journal Submitted to: Embo Molecular Medicine

Manuscript Number: EMM-2020-13366-V2